# The influence of dynein processivity control, MAPs, and microtubule ends on directional movement of a localising mRNA

Harish Chandra Soundararajan[†], Simon L Bullock*

Division of Cell Biology, MRC Laboratory of Molecular Biology, Cambridge, United Kingdom

**Abstract** Many cellular constituents travel along microtubules in association with multiple copies of motor proteins. How the activity of these motors is regulated during cargo sorting is poorly understood. In this study, we address this issue using a novel in vitro assay for the motility of localising *Drosophila* mRNAs bound to native dynein-dynactin complexes. High precision tracking reveals that individual RNPs within a population undergo either diffusive, or highly processive, minus end-directed movements along microtubules. RNA localisation signals stimulate the processive movements, with regulation of dynein-dynactin's activity rather than its total copy number per RNP, responsible for this effect. Our data support a novel mechanism for multi-motor translocation based on the regulation of dynein processivity by discrete cargo-associated features. Studying the in vitro responses of RNPs to microtubule-associated proteins (MAPs) and microtubule ends provides insights into how an RNA population could navigate the cytoskeletal network and become anchored at its destination in cells.

*For correspondence: sbullock@mrc-lmb.cam.ac.uk

**Present address:** †Wyss Institute for Biologically Inspired Engineering, Harvard University, Boston, United States

**Competing interests:** The authors declare that no competing interests exist.

**Reviewing editor**: Randy Schekman, Howard Hughes Medical Institute, University of California, Berkeley, United States

## Introduction

Microtubule-based motility plays a major role in the distribution and sorting of organelles, vesicles, and macromolecules within cells. The significance of this process is underscored by the association of mutations in microtubule motor proteins and their co-factors with several neurological disorders (*Hirokawa et al., 2010*; *Schiavo et al., 2013*). Furthermore, many pathogens exploit cellular microtubule motors during infection (*Grieshaber et al., 2003*; *Greber and Way, 2006*; *Ramsden et al., 2007*). Despite the fundamental importance of the process, the mechanisms by which cargos are trafficked along microtubules are poorly understood.

There is substantial evidence that individual cargos simultaneously associate with multiple microtubule motors (*Gross et al., 2007*). Not only can a single cargo associate with several copies of the same kind of motor (e.g., *Leopold et al., 1992*; *Welte et al., 1998*; *Hendricks et al., 2010*; *Encalada et al., 2011*; *Rai et al., 2013*), but opposite polarity motors are often stably bound (*Ling et al., 2004*; *Pilling et al., 2006*; *Shubeita et al., 2008*; *Soppina et al., 2009*; *Hendricks et al., 2010*; *Encalada et al., 2011*). Thus, in order to understand cargo trafficking in vivo it is essential to learn how the activity of multiple motors is orchestrated.

The mechanisms governing translocation of cargos by teams of microtubule motors have predominantly been tackled in two ways. One set of studies has analysed motility of cargo populations within cells (e.g., *Kural et al., 2005*; *Shubeita et al., 2008*; *Reis et al., 2012*; *Rai et al., 2013*). Although physiologically relevant, the mechanistic insights that can be derived from this approach are limited by the complex in vivo environment. The second set of studies have analysed the in vitro behaviours of

**eLife digest** For a cell to do its job, the different components inside it need to be moved to different locations. This is achieved by an elaborate cellular transport system. To move a component to where it needs to be, motor proteins bind to it, often with the assistance of other 'accessory' proteins. This cargo-motor complex then moves along a network of tracks within the cell. Viruses also exploit this transport system in order to be trafficked to specific parts of the cell during their life cycles.

Many cargos are moved along microtubule tracks. Multiple microtubule motor proteins often attach to the same cargo, but it is unclear how they work together during transport. Previous studies have attempted to address this issue by attaching motor proteins to artificial cargoes, such as synthetic beads. However, these experiments did not include some of the accessory proteins that are thought to play a role during transport within the living cell.

Soundararajan and Bullock have now examined how complexes containing multiple motors bound to accessory proteins move molecules of messenger RNA to specific sites within cells. By visualising fruit fly mRNA moving along microtubules attached to a glass surface, the transport process can be studied in detail. It appears that the complexes travel using one of two methods: they either diffuse along the microtubules, which they can do in either direction, or they power themselves along the microtubules, which they can only do in one direction. Although previous experiments with artificial cargos suggested that the number of motors in the complex determines the likelihood of one-way traffic, it appears that one or more accessory proteins are actually in control during mRNA transport.

Soundararajan and Bullock also documented how the mRNA-motor complexes react to roadblocks and dead-ends on the microtubule highway. Rather than letting go of the microtubule upon such an encounter, the complexes can reverse back down the track. This behaviour may help them to find a new route to their destination.

artificial cargos, such as beads or DNA origami, coupled to isolated motor proteins or motor domains (e.g., *Vale et al., 1985*; *Block et al., 1990*; *Mallik et al., 2005*; *Diehl et al., 2006*; *Ross et al., 2006*; *Vershinin et al., 2007*; *Derr et al., 2012*; *Furuta et al., 2013*). These in vitro studies have provided evidence that small increases in the number of purified motors of the same polarity strongly augment the average travel distance in that direction (*Block et al., 1990*; *Mallik et al., 2005*; *Vershinin et al., 2007*; *Derr et al., 2012*; *Furuta et al., 2013*). However, these experiments did not include potential regulatory co-factors that associate with motors in vivo. Thus, substantial debate persists over whether net movement of physiological cargo-motor complexes is dominated by motor copy number or by higher order mechanisms that regulate motor activity (*Gross, 2004*; *Kural et al., 2005*; *Shubeita et al., 2008*; *Elting and Spudich, 2012*; *Reis et al., 2012*).

Another important unresolved issue is how, when navigating to their destination in vivo, cargo-motor complexes cope with other proteins that decorate microtubules. In vitro experiments have shown that the plus end-directed kinesin-1 motor frequently detaches when encountering microtubule-associated proteins (MAPs) (*Vershinin et al., 2007*; *Dixit et al., 2008*; *Telley et al., 2009*; *McVicker et al., 2011*). In contrast, upon such an obstacle encounter, individual bidirectional dynein-dynactins often remain attached to a microtubule and undergo a reversal in travel direction (*Dixit et al., 2008*). It remains to be tested in a defined system how intact transport complexes, containing cargo, multiple motors, and potential regulatory proteins, react to obstacles on their tracks. It is also not known how these complexes respond when they reach the ends of the microtubules.

Each of these problems is exemplified by the trafficking of developmentally important mRNAs in the syncytial *Drosophila* embryo. Cytoplasmic injection of in vitro synthesised fluorescent transcripts has shed light on the mechanisms governing RNA sorting in this system. These experiments have provided evidence that apical mRNA localisation is achieved by a bidirectional translocation process in which, on average, minus end-directed transport by the multi-subunit dynein motor and its large accessory complex dynactin predominates (*Wilkie and Davis, 2001*; *Bullock et al., 2006*; *Vendra et al., 2007*). Upon reaching the apical cytoplasm, the ribonucleoprotein complexes (RNPs) are statically anchored by an unknown, dynein-dependent mechanism (*Delanoue and Davis, 2005*). mRNAs that

are uniformly distributed also move bidirectionally, but with little net directional bias (*Bullock et al., 2006*; *Amrute-Nayak and Bullock, 2012*). Intriguingly, dynein-dynactin is required for both plus end- and minus end-directed motion of the localising and uniformly distributed RNPs formed upon injection (*Bullock et al., 2006*; *Vendra et al., 2007*). Dynein is also needed for efficient spreading of uniformly distributed endogenous RNAs from the perinuclear region, supporting a physiological requirement for the motor complex in bidirectional RNA motion (*Bullock et al., 2006*). These findings, together with the failure to detect functional evidence for the involvement of a kinesin motor (*Vendra et al., 2007*), suggest that plus end movements of RNPs are driven by dynein moving in this direction, a property that has been documented in several in vitro studies of the motor (*Schliwa et al., 1991*; *Wang et al., 1995*; *Wang and Sheetz, 2000*; *Mallik et al., 2005*; *Ross et al., 2006*; *Miura et al., 2010*; *Walter et al., 2012*).

Net minus end transport of apical transcripts is dependent on RNA localisation signals, which are comprised of specialised stem-loops that recruit additional dynein-dynactin complexes to RNPs through the Egalitarian (Egl) and Bicaudal-D (BicD) adaptor proteins (*Bullock et al., 2006*; *Dienstbier et al., 2009*; *Amrute-Nayak and Bullock, 2012*). Egl binds directly to the localisation signals (*Dienstbier et al., 2009*) and the light chain subunit of dynein (*Navarro et al., 2004*), whereas BicD interacts simultaneously with Egl (*Navarro et al., 2004*; *Dienstbier et al., 2009*) and multiple sites in the dynein-dynactin complex (*Hoogenraad et al., 2001*; *Splinter et al., 2012*). Egl and BicD do not appear to contribute to the binding of the dynein-dynactin complex to RNA at sites other than localisation signals (*Bullock et al., 2006*; *Dix et al., 2013*), and the RNA features and protein factors that fulfil this task have not been identified.

Recent proteomic work by our group (*Dix et al., 2013*) has shown that Lissencephaly-1 (Lis1) is also a component of dynein-dynactin complexes associated with localising and uniformly distributed RNAs. Lis1 promotes the recruitment of dynein-dynactin to RNAs (*Dix et al., 2013*) and may also regulate mechanochemistry of the cargo-associated motor (*McKenney et al., 2010*; *Huang et al., 2012*; *Vallee et al., 2012*). The study of Dix et al. supported the existence of a core functional complex recruited to localisation signals, consisting of Egl, BicD, dynein-dynactin, and Lis1 (*Dix et al., 2013*). However, it is not known whether the dynein-dynactin recruited in this manner is more likely to engage in minus end-directed motion than that recruited elsewhere in the RNA. Alternatively, the localisation signals could drive net minus end motion simply by recruiting more copies of functionally equivalent dynein-dynactin complexes per RNP.

In order to begin to address these mechanistic issues, we have developed a novel in vitro RNA motility assay that combines the manipulability of a cell-free system with the physiological relevance of cargo-motor complexes assembled from *Drosophila* embryo extract. We have used the unique advantages of this system to examine the mechanism of directionally biased motility by multi-motor assemblies, the response to potential obstacles, and the consequences of reaching microtubule ends.

## Results

### A novel in vitro assay to study mRNA motility

Optical limitations of the *Drosophila* embryo preclude direct visualisation of RNPs moving along individual microtubules. We therefore previously established an in vitro assay for mRNA motility by capturing RNA-motor complexes from embryo extract through the affinity of motors for microtubules (*Amrute-Nayak and Bullock, 2012*). However, detailed characterisation of RNA motility was challenging due to the limited number of motile RNPs in each imaging chamber. We therefore developed RAT-TRAP (RNA Transport After Tethered RNA Affinity Purification), a method to efficiently study RNP motility in an in vitro setting.

RNA-motor complexes were assembled by incubating embryo extract with in vitro transcribed RNAs immobilised on streptavidin-coated beads via a streptavidin-binding RNA aptamer (*Figure 1A*). The RNAs were labelled by the stochastic incorporation of fluorophore-coupled UTP during the transcription reaction, permitting downstream visualisation by fluorescence microscopy. The assembled RNA-motor complexes were washed briefly and eluted from the beads with biotin, which competes for the interaction of the aptamer with streptavidin. This methodology leads to the recruitment of dynein-dynactin complexes to RNA localisation signals in association with Egl, BicD, and Lis1 (*Dix et al., 2013*). To assay in vitro RNP motility, the eluate was supplemented with a saturating level of ATP and added to a chamber containing polarity-marked, GmpCpp-stabilised microtubules bound to a coverslip (*Figure 1A*). RNAs and microtubules were then visualised with total internal reflection (TIR) microscopy.

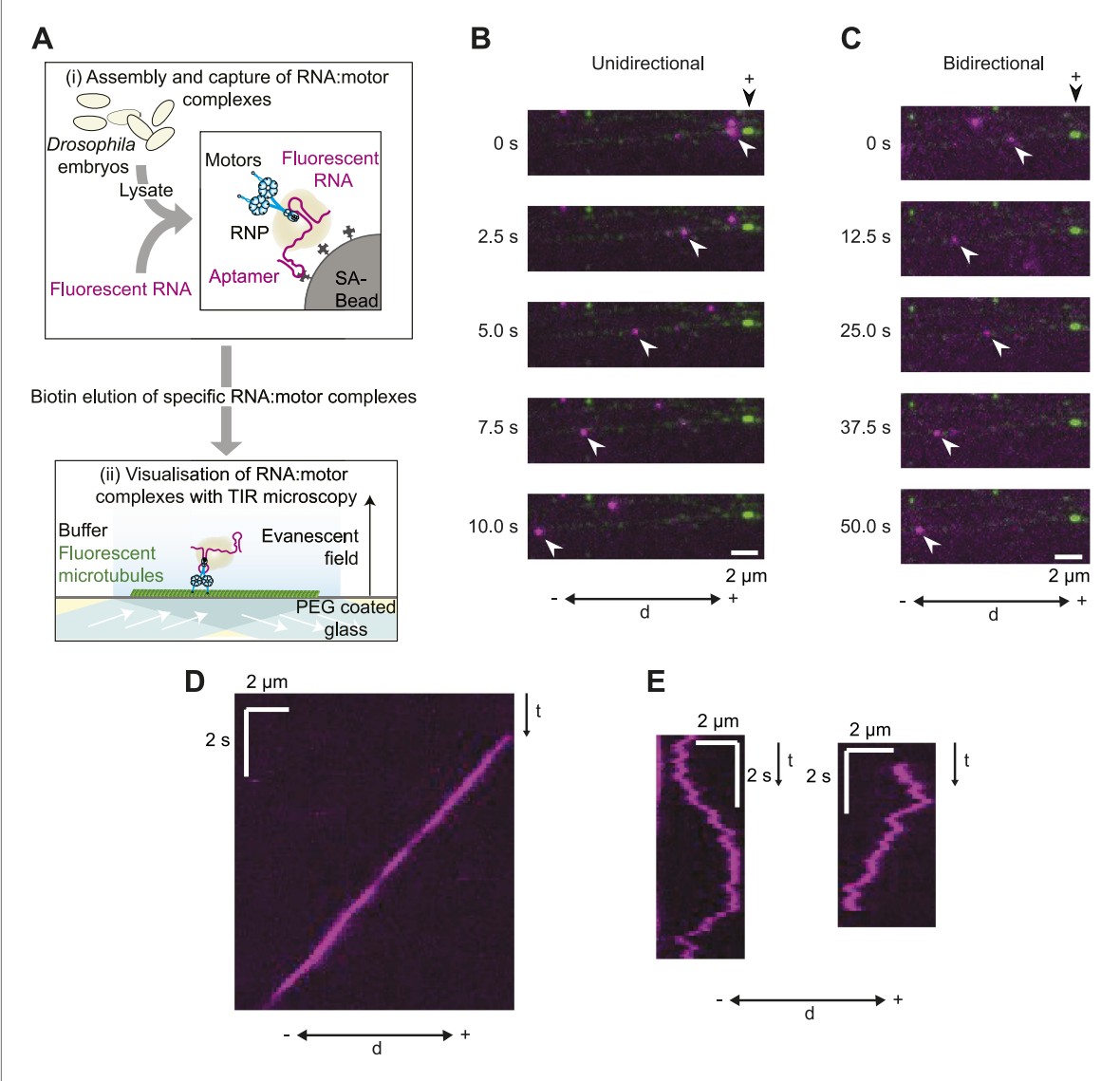

**Figure 1**. A novel in vitro assay to study mRNA motility. (**A**) Schematic of assay; see text for details. SA, streptavidin; Aptamer, streptavidin-binding RNA aptamer. (**B** and **C**) Stills generated from time-lapse series of motile unidirectional (**B**) and bidirectional (**C**) Cy3-*h* wild-type (*h*^WT) RNPs (magenta). Double-headed arrow indicates orientation of microtubule (− and +, minus and plus end); plus end (black arrowhead above top still) is marked by greater incorporation of HiLyte 647-tubulin (green). (**D** and **E**) Kymographs (time-distance plots) of examples of unidirectional (**D**) and bidirectional (**E**) RNPs; t, time.

The following figure supplements are available for figure 1:

**Figure supplement 1**. Assessing RNA copy number in *h*^WT RNPs and tracking accuracy in the RAT-TRAP assay.

Our initial experiments focused on the well-characterised *hairy* (*h*) mRNA, which requires a 124 nt localisation signal (the *h* localisation element [*HLE*]) in its ~800 nt 3′-untranslated region (UTR) to localise apically in the *Drosophila* embryo (***Bullock et al., 2003***). Embryo extracts were incubated with immobilised *h* wild-type 3′ UTR RNAs (*h*^WT), which contained an average of ~8 Cy3 dyes per molecule. 50–80 microtubule-associated RNPs were typically observed following injection of the eluate into an imaging chamber, with ~60–70% of these displaying motility during the ~100 s of data acquisition. There was a strong overall minus end bias to motility of the *h*^WT RNP population, as is the case in vivo (***Bullock et al., 2003***, ***2006***). 24 ± 1% of motile RNPs analysed per chamber moved exclusively unidirectionally towards the minus ends of microtubules (mean ± SEM, 10 chambers; e.g., ***Figure 1B,D***; ***Video 1***).

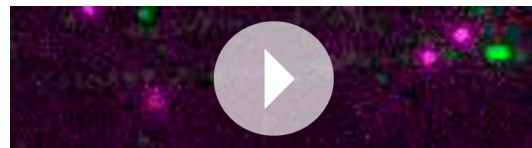

**Video 1**. Example of unidirectional motion of a Cy3-$h^{WT}$ RNP towards the minus end of a polarity-marked microtubule. RNA, pseudocoloured magenta; microtubule, pseudocoloured green. Microtubule plus end is labelled with a highly fluorescent segment. This RNP pauses upon reaching the minus end of the microtubule; we also observed instances of dissociation of unidirectional RNPs upon reaching the minus end (*Figure 6D,E*). Video corresponds to 12.4 s; width of the image is 16.7 μm.

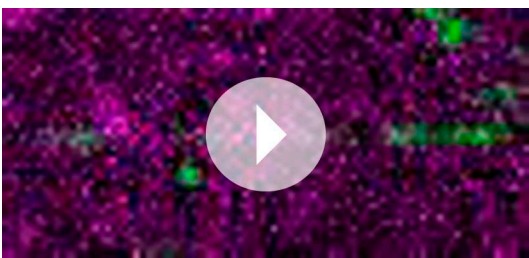

**Video 2**. Example of bidirectional motion of a Cy3-$h^{WT}$ RNP on an individual microtubule. RNA, pseudocoloured magenta; microtubule, pseudocoloured green. Microtubule plus end located outside of field-of-view (towards the right). Video corresponds to 62.5 s; width of the image is 9.6 μm.

These movements were highly processive, often ending when RNPs paused or detached upon reaching the microtubule minus end. The remainder of motile $h^{WT}$ RNPs moved bidirectionally, with frequent switches in direction (e.g., *Figure 1C,E*; *Video 2*) and no overt directional bias at the population level (see below for quantification).

We have recently shown that, unlike the large RNPs formed after injection of fluorescent RNA into the embryo, endogenous $h$ RNPs contain a single copy of the RNA molecule (*Amrute-Nayak and Bullock, 2012*). To assess the copy number of $h$ RNA per RNP in the RAT-TRAP assay, exclusively Alexa488-labelled $h^{WT}$ RNAs and exclusively Cy3-labelled $h^{WT}$ RNAs were incubated simultaneously with the beads before the addition of extract. All motile RNPs assembled in such experiments contained only one type of fluorophore (*Figure 1—figure supplement 1A,B*). These data indicate that, as was the case for our previous in vitro assay for RNA motility (*Amrute-Nayak and Bullock, 2012*), the RAT-TRAP assay reports on the motility of RNPs containing a single fluorescent RNA molecule.

The number of motile $h^{WT}$ RNPs observed in RAT-TRAP assays was ~10-fold greater than that observed using our previous method ([*Amrute-Nayak and Bullock, 2012*] and data not shown). Furthermore, functionalisation of glass surfaces with polyethylene glycol greatly reduced non-specific binding of fluorescent RNPs to the cover-slip in the new assay and thereby allowed the implementation of automatic, sub-pixel tracking of microtubule-associated complexes (*Figure 1—figure supplement 1C,D*) with high temporal precision (15 frames per second [fps]). Our previous assay (*Amrute-Nayak and Bullock, 2012*) used

manual analysis of kymographs generated from images captured at a much lower frame rate (~3 fps). Thus, the RAT-TRAP assay constitutes a substantial step forward in the ability to analyse the movement of RNPs in vitro.

## Characterisation of the motile properties of $h^{WT}$ RNPs

To investigate the global properties of unidirectional and bidirectional motion, we analysed the relationship of mean square displacement (MSD) of tracked RNP trajectories with time. As expected, unidirectional $h^{WT}$ RNP trajectories showed a quadratic dependence of mean square displacement (MSD) over time (average slope of ~2.0 in a log–log plot [*Figure 2A*]; see *Figure 2—figure supplement 1A,C* for additional MSD analysis). This observation indicates an underlying active transport process (*Saxton, 1997*; *Sanchez et al., 2012*). In contrast, the relationship of MSD of bidirectional RNPs over time was linear (average slope of ~1.0 in the log–log plot (*Figure 2A*; *Figure 2—figure supplement 1B,C*). Thus, there appears to be a strong diffusive component to the movement of bidirectional $h^{WT}$ RNPs (*Saxton, 1997*; *Sanchez et al., 2012*).

We next extracted runs from RNP trajectories and analysed their length and velocity. A run was operationally defined as a bout of uninterrupted motion in one direction before a reversal, pause, detachment from the microtubule or curtailment by the end of image acquisition. The mean length and velocity of individual runs of unidirectional $h^{WT}$ RNPs were ~4 μm and ~1.2 μm.s$^{-1}$, respectively (*Figure 2—figure supplement 2A,B*). The mean velocity of individual runs of bidirectional $h^{WT}$ RNPs in both directions (~0.8 μm.s$^{-1}$) was similar to that of the unidirectional counterparts, whereas the mean

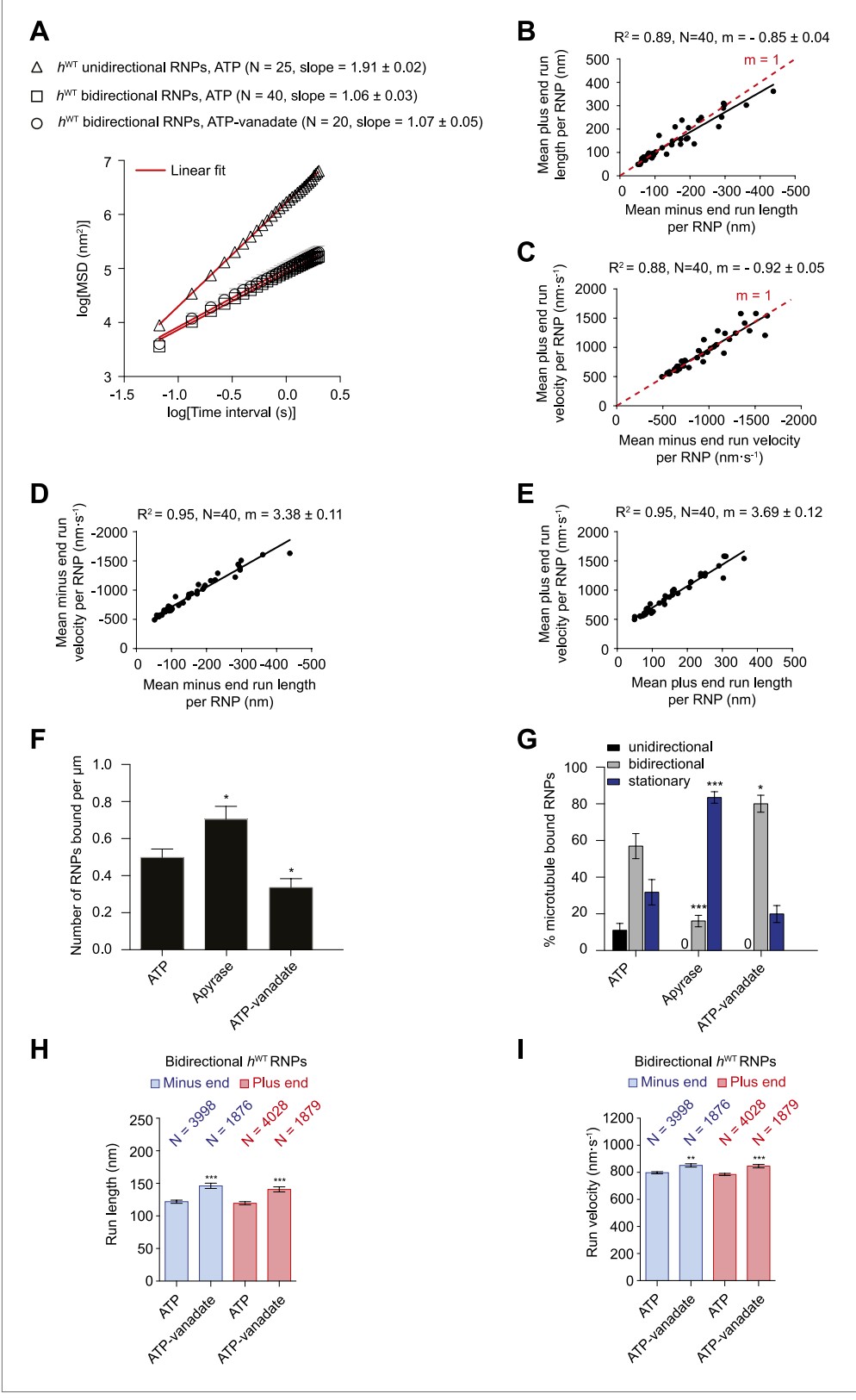

**Figure 2**. Characterisation of the motile properties of $h^{WT}$ RNPs. (**A**) Mean square displacement (MSD) of $h^{WT}$ RNP trajectories as a function of time plotted in a log–log format. Mean slopes (± SEM) were calculated from a linear fit to the data. Plot of MSD vs time on non-logarithmic axes is shown in ***Figure 2—figure supplement 1A,B***. ATP, 2.5

*Figure 2. Continued on next page*

*Figure 2. Continued*

mM ATP; ATP-vanadate, 2.5 mM ATP plus 100 μM vanadate. N, number of RNPs analysed. See *Figure 2—figure supplement 1C* for slopes of log–log MSD(t) for individual RNPs in each population. (**B**–**E**) Correlation analysis of mean run length and run velocity for the bidirectional subset of $h^{WT}$ RNPs. Strong correlations exist for individual RNPs between (**B**) mean minus end and mean plus end run length, (**C**) mean minus end and mean plus end velocity, (**D**) mean minus end run length and mean minus end velocity, and (**E**) mean plus end run length and mean plus end velocity. Only bidirectional RNPs with ≥20 runs in total were used for these analyses (note that no such cut-off was applied for the analysis in **H** and **I**). $R^2$, correlation coefficient; N, number of RNPs analysed; m, slope. There is no significant bias in minus end vs plus end motile properties in **B** and **C** (red line represents slope expected for no bias). (**F**) Mean number of $h^{WT}$ RNPs bound per μm of microtubule per movie in the presence of 2.5 mM ATP, 20 U·ml$^{-1}$ apyrase and 2.5 mM ATP plus 100 μM vanadate (ATP-vanadate). Means were calculate from values for 12 microtubules selected at random in at least three imaging chambers. (**G**) Mean percentage of microtubule-associated $h^{WT}$ RNPs that were unidirectional, bidirectional, or stationary. Means were calculated from 12 microtubules as in **F**. (**H** and **I**) Mean run lengths (**H**) and velocities (**I**) of individual runs of bidirectional $h^{WT}$ RNPs. N, number of individual runs of RNPs (number of RNPs from which the individual runs were extracted was 40 for ATP and 20 for ATP-vanadate). See *Figure 2—figure supplement 2C,D* for distribution of run lengths and velocities. Errors represent SEM in all panels. In **F–I**, ***p<0.001; **p<0.01; *p<0.05, compared to the ATP condition for the same parameter (Mann–Whitney non-parametric *t* test). Images for all analyses in the figure were acquired at 15 fps.
The following figure supplements are available for figure 2:

**Figure supplement 1**. Supplementary MSD analysis of $h^{WT}$ RNP trajectories.

**Figure supplement 2**. Distributions of run lengths and velocities of unidirectional and bidirectional $h^{WT}$ RNPs.

run lengths were much shorter (*Figure 2—figure supplement 2C,D*). In both directions the mean length of individual runs of bidirectional RNPs was ~120 nm; only a small fraction of runs were longer than 500 nm, with the longest ~2 μm (*Figure 2—figure supplement 2C* and accompanying legend). These run lengths are much shorter than those previously documented for force-generating dynein and kinesin motors bound to other bidirectional cargos (*Shubeita et al., 2008*; *Soppina et al., 2009*; *Schuster et al., 2011*; *Reis et al., 2012*; *Rai et al., 2013*).

Analysis of the mean run length and velocities of individual bidirectional $h^{WT}$ RNPs revealed a great deal of variation for each of these values per RNP (*Figure 2B–E*). This presumably reflects compositional heterogeneity, a feature that has been observed in populations of cargo-motor complexes in vivo (*Encalada et al., 2011*; *Schuster et al., 2011*). Despite their variation across the population, the quantitative parameters of motion were highly correlated to each other for individual RNPs (*Figure 2B–E*). The mean minus end and plus end run lengths were very similar for any given RNP (*Figure 2B*). Individual RNPs also exhibited mean velocities that were highly similar in both the minus end and plus end direction (*Figure 2C*). Mean run lengths and run velocities for individual RNPs were also highly correlated (*Figure 2D,E*). Thus, complexes that move faster on average tended to move further before a run ends. Overall, our correlation analysis reveals a high degree of coupling between motile properties in each direction for individual RNPs.

We next studied the effects of alternate nucleotide states on the motility of RNPs. Depletion of ATP and ADP with apyrase (*Higuchi et al., 1997*; *Ma and Taylor, 1997*) led to a significant increase in the number of microtubule-associated RNPs compared to the ATP condition (*Figure 2F*). In the presence of apyrase there was also a large increase in the proportion of microtubule-bound RNPs that were stationary, with unidirectional behaviour absent and a strong reduction in the proportion of bidirectional complexes (*Figure 2G*). The affect of apyrase on unidirectional motion is entirely expected as processive movement of dynein is dependent on ATP hydrolysis (*Roberts et al., 2013*). The inhibition of bidirectional motion by apyrase could conceivably be explained by dynein normally stepping towards both the minus and plus end of microtubules through ATP hydrolysis—as has been documented in two previous studies of purified motor complexes (*Ross et al., 2006*; *Walter et al., 2012*)—with frequent reversals between bouts of motion in each direction. Alternatively, RNPs could undergo passive diffusion along microtubules in the presence of ATP, with the presence of the motor on RNPs stimulating long-term arrest in the absence of ATP due to its tight microtubule binding in the no nucleotide state (*Roberts et al., 2013*). Both these scenarios, or a combination of the two, could account for the relatively short

mean run lengths, diffusive MSD properties, and the tight coupling of minus and plus end motile properties for individual bidirectional RNPs in the presence of ATP.

To discriminate between these possibilities we performed motility assays in the presence of both ATP and vanadate. Vanadate inhibits the ATPase activity of dynein and mimics the ADP·Pi state, which is associated with weak affinity for microtubules (*Shimizu and Johnson, 1983*; *Miura et al., 2010*). There was a partial reduction in the number of RNPs bound to microtubules in the presence of vanadate (*Figure 2F*). As expected, vanadate abolished unidirectional motion of the microtubule-associated RNPs (*Figure 2G*). Tellingly, the percentage of microtubule-associated RNPs that underwent bidirectional motion was not decreased in the presence of ATP-vanadate compared to ATP alone (*Figure 2G*). In fact, in the presence of vanadate there was an increase in the proportion of microtubule-associated RNPs that were bidirectional, possibly due to a contribution of RNPs that would otherwise be unidirectional undergoing switching to a bidirectional state.

The relationship of MSD vs time was very similar for the bidirectional RNPs in the presence of ATP-vanadate and ATP (*Figure 2A*, *Figure 2—figure supplement 1B,C*). Similar distributions of lengths and velocities or individual runs in both the minus and plus end direction were also observed for bidirectional RNPs in both conditions (*Figure 2—figure supplement 2C,D*). Thus, even the occurrence of relatively long runs of $h^{WT}$ RNPs was not prevented by vanadate (*Figure 2—figure supplement 2C,D*). Vanadate actually caused a subtle but statistically significant increase in the mean run length and velocity of bidirectional RNPs in the minus and plus end directions (*Figure 2H,I*), implying a modulation of the diffusive properties of RNPs when dynein is in the ADP.Pi state. Overall, our results demonstrate that an active energy transduction property of dynein is required for unidirectional, minus end-directed RNP motion but not for motion of bidirectional RNPs in either the minus or plus end direction. Thus, individual $h^{WT}$ RNPs can adopt two discrete behaviours in vitro: unidirectional motion, driven by processive movement of dynein that is associated with ATP hydrolysis and bidirectional motion that appears to be dominated by passive diffusion.

## Increasing dynein-dynactin numbers through RNA localisation signals promotes unidirectional, minus end-directed transport

As described in the introduction, there is substantial debate on the influence of total motor copy number on the translocation of cargos. To assess the consequences of copy number of native motor complexes on the motion of a physiological cargo in vitro we increased or decreased the number of native dynein-dynactin complexes per $h$ RNP. This was achieved by manipulating the *HLE* RNA sequences that are naturally responsible for their recruitment. The key feature of the *HLE* is stem-loop 1 (SL1) (*Bullock et al., 2003*), which recruits dynein-dynactin through the adaptor proteins Egl and BicD (*Dienstbier et al., 2009*; *Dix et al., 2013*). We therefore generated Cy3-labelled RNAs that contained fusions of the streptavidin-binding aptamer to a $h$ 3′UTR in which the entire *HLE* is replaced by three copies of the SL1 element ($h^{SL1×3}$) or a heterologous sequence from the glutathione-S-transferase (GST) RNA ($h^{\Delta LE}$) (*Figure 3A*).

We used stepwise GFP photobleaching to confirm that the number of dynein and dynactin subunits associated with individual RNPs increased with increasing numbers of SL1 elements per RNA molecule. Photobleaching analysis was performed on motor complexes that were assembled on the $h$ RNA variants using embryo extracts expressing GFP-tagged versions of Dynein light intermediate chain (GFP::Dlic) or the dynactin subunit p50-Dynamitin (GFP::Dmn) and stably bound to microtubules in the absence of nucleotide (*Amrute-Nayak and Bullock, 2012*) (*Figure 3A–C*, *Figure 3—figure supplement 1A–C*).

Previous experiments had shown that GFP::Dlic is incorporated into dynein-dynactin complexes in accordance with its abundance in embryo extract relative to endogenous Dlic (*Amrute-Nayak and Bullock, 2012*). Thus, by using fluorescent immunoblotting to determine the proportion of total Dlic in the extract that was labelled with GFP (*Figure 3—figure supplement 1D*), we could approximate the mean number of Dlic copies on each of the $h$ RNA variants (*Figure 3—figure supplement 1E*). The estimated means ± SEM of Dlic copies for $h^{WT}$, $h^{\Delta LE}$, and $h^{SL1×3}$ RNPs were, respectively, 6.73 ± 0.44, 4.36 ± 0.33, and 13.85 ± 0.67. It has been reported that there are two Dlic molecules per dynein motor (*King et al., 1996*; *Trokter et al., 2012*) and hence our estimates of Dlic copy number on $h$ RNA variants are consistent with 3.36 ± 0.21, 2.18 ± 0.16, and 6.92 ± 0.33 dynein motors associated with $h^{WT}$, $h^{\Delta LE}$, or $h^{SL1×3}$ RNPs, respectively. We wish to stress that these values are only estimates of the average dynein copy number on each RNA due to limitations of the photobleaching technique in determining

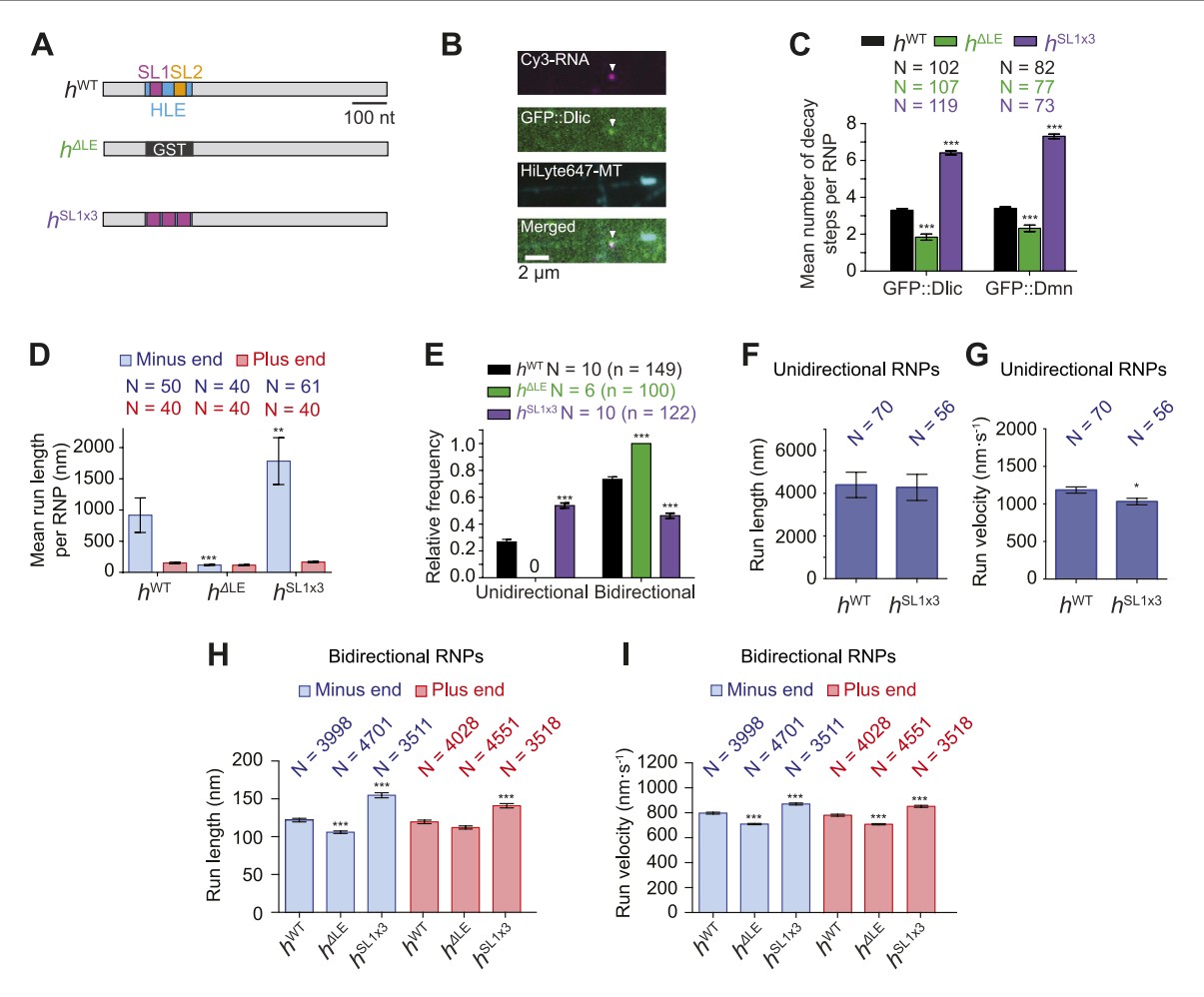

**Figure 3**. Manipulating the copy number of native dynein–dynactin complexes on individual RNPs. (**A**) Schematic of *h* RNA variants fused to streptavidin aptamers and used in this study. *h*^ΔLE and *h*^SL1x3 have replacements of the 124 nt HLE region with, respectively, a heterologous sequence from the Glutathione-S-transferase (GST) gene and three copies of stem-loop 1 (SL1) separated by short spacers. (**B**) Example of an RNP in vitro containing Cy3-*h* RNA (magenta) and GFP::Dlic (green), which is immobilised on a polarity-marked microtubule (cyan) by the omission of ATP. (**C**) Stepwise GFP photobleaching analysis of RNPs assembled on *h* RNA variants. Note significant relative changes in the copy number of Dlic and Dmn upon increasing or decreasing copy number of SL1. N, number of photobleaching traces analysed. See *Figure 3—figure supplement 1B* for distribution of values. (**D**) Mean run length per RNP for each RNA species. N, number of RNPs analysed. (**E**) Mean proportion of motile RNPs per imaging chamber that are unidirectional or bidirectional. No unidirectional RNPs were observed for *h*^ΔLE. N, number of chambers analysed; n, total number of RNPs. (**F** and **G**) Mean length (**F**) and velocity (**G**) of individual runs of unidirectional RNPs. N, number of runs (from 25 and 20 RNPs for *h*^WT and *h*^SL1x3, respectively [many RNPs have more than one run due to interruptions of bouts of minus end-directed motility by short-lived pauses]). (**H** and **I**) Mean length (**H**) and velocity (**I**) of individual runs of bidirectional RNPs. N, number of runs (from 40 RNPs each for *h*^WT, *h*^ΔLE, and *h*^SL1x3). In **D–I**, all experiments were performed in 2.5 mM ATP; error bars represent SEM. ***p<0.001; **p<0.01; *p<0.05, compared to *h*^WT values for the same parameter (Mann–Whitney non-parametric *t* test).

The following figure supplements are available for figure 3:

**Figure supplement 1**. Supplemental data on GFP photobleaching analysis of relative dynein and dynactin copy number per *h*^WT, *h*^ΔLE, or *h*^SL1x3 RNP.

**Figure supplement 2**. Distributions of run lengths and velocities of *h*^SL1x3 and *h*^ΔLE RNPs.

precise copy number of fluorescent molecules (*Das et al., 2007*; *Shu et al., 2007*). Nonetheless, our analysis clearly reveals that manipulating SL1 copy number in the different *h* RNA variants causes significant changes, in relative terms, in the average copy number of dynein and dynactin complexes per RNP.

We next analysed the motile behaviour of the different *h* RNA variants in the RAT-TRAP assay in the presence of ATP. The population of *h*^SL1x3 RNPs had a significantly greater minus end-directed bias than

the $h^{WT}$ RNPs (*Figure 3D*). This was associated with a doubling in the proportion of motile $h^{SL1x3}$ RNPs undergoing exclusively unidirectional movement towards the minus end of microtubules compared to $h^{WT}$ RNPs (*Figure 3E*), without substantially altering the mean run lengths and velocities of these events (*Figure 3F,G*, *Figure 3—figure supplement 2A,B*). Reducing the copy number of dynein-dynactin per RNP by using the $h^{\Delta LE}$ RNA resulted in exclusively bidirectional motion (*Figure 3E*) with no significant net directional bias (*Figure 3D*). Thus, there was a correlation between the probability of unidirectional minus end-directed motion of RNPs and the average number of associated dynein-dynactin complexes.

The distribution of run lengths and velocities of the bidirectional RNPs in the minus and plus end directions were similar for $h^{WT}$, $h^{SL1x3}$, and $h^{\Delta LE}$ RNAs (*Figure 3H,I*, *Figure 3—figure supplement 2C,D*). There was, however, a subtle increase in both minus and plus end mean run lengths and velocities with increasing numbers of SL1 elements (*Figure 3H,I*, *Figure 3—figure supplement 2C,D*). These data suggest that increasing dynein-dynactin copy number through localisation signals does not have a major influence on the motile properties of bidirectional RNPs. However, the correlation between increased dynein-dynactin number and increased lengths and velocities of runs in both the minus and plus end direction is consistent with this complex playing a direct role in mediating bidirectional movement of RNPs along microtubules.

## The *HLE* alone promotes unidirectional motion

The experiments described above with the $h^{\Delta LE}$, $h^{WT}$, and $h^{SL1x3}$ RNAs show that the major mechanism through which increasing the number of RNA localisation signals promotes net minus end-directed motion is by increasing the probability of long, unidirectional transport in this direction. Consistent with previous observations (*Bullock et al., 2006*; *Amrute-Nayak and Bullock, 2012*; *Dix et al., 2013*), we find that RNAs lacking localisation signals still recruit dynein-dynactin but move exclusively bidirectionally (*Figure 3C,E*). Thus, localisation signals may promote unidirectional motion simply by increasing the average copy number of dynein-dynactin complexes bound per RNP. Such a model is consistent with previous work showing that individual purified dynein-dynactin complexes move bidirectionally in vitro, with additional dynein-dynactin complexes bound to a bead driving unidirectional motion in the minus end direction (*Ross et al., 2006*). Alternatively, dynein-dynactins recruited to localisation signals may be more likely to undergo unidirectional motion than those bound elsewhere in the RNA.

To discriminate between these possibilities we performed RAT-TRAP assays with Cy3-labelled *HLE* RNA, which lacks the dynein-dynactin binding sites provided by the rest of the *h* 3'UTR sequences. Importantly, we first confirmed using stepwise photobleaching with GFP-Dlic extracts that there is a significant reduction in the relative copy number of the motor complex on *HLE* RNPs compared to $h^{WT}$ RNPs (*Figure 4A*, *Figure 4—figure supplement 1A,B*). The average number of GFP::Dlic photobleaching steps on the *HLE* was in fact statistically indistinguishable from that observed for $h^{\Delta LE}$ (2.21 ± 0.15 and 2.14 ± 0.16, respectively; *Figure 4A* and *Figure 3C*).

If the total copy number of the dynein complex is the sole determinant of unidirectional transport, one would expect *HLE* RNPs to be less likely than $h^{WT}$ RNPs to exhibit this behaviour, possibly exhibiting exclusively bidirectional motion as is the case for $h^{\Delta LE}$. However, this was not the case. Approximately 65% of RNPs containing the *HLE* RNA exhibited unidirectional motion towards the minus end of microtubules, compared to 24% of those containing the $h^{WT}$ RNA (*Figure 4B*). Consequently, the *HLE* RNP population had a particularly strong minus end directional bias (*Figure 4C*). These data indicate that total dynein-dynactin copy number is not the primary determinant of unidirectional RNP movement. Instead the *HLE* promotes the ability of the associated dynein-dynactin to undergo unidirectional transport, with features in the rest of the 3'UTR seemingly antagonising this effect.

The mean run lengths and velocities of unidirectional runs of *HLE* RNPs were indistinguishable from those of $h^{WT}$ RNPs ((*Figure 4D,E*, *Figure 4—figure supplement 1C,D*); note that in this series of experiments, and others involving the *HLE*, the image exposure time was increased (acquisition rate, 4.2 fps) to compensate for the reduced number of Cy3 dyes incorporated into the shorter RNA. Because more pauses and reversals are missed at the lower frame rate, there are differences in the absolute values of measured lengths and velocities of runs for the $h^{WT}$ RNPs compared to analysis at 15 fps, *Figure 4* legends). The comparison of unidirectional runs of *HLE* and $h^{WT}$ RNPs, together with the earlier evaluation of unidirectional motility of $h^{SL1x3}$ and $h^{WT}$ RNPs (*Figure 3F,G*), demonstrates that once initiated the unidirectional mode of transport is not strongly influenced by the average number of cargo-associated dynein-dynactins. The bidirectional *HLE* RNPs exhibited

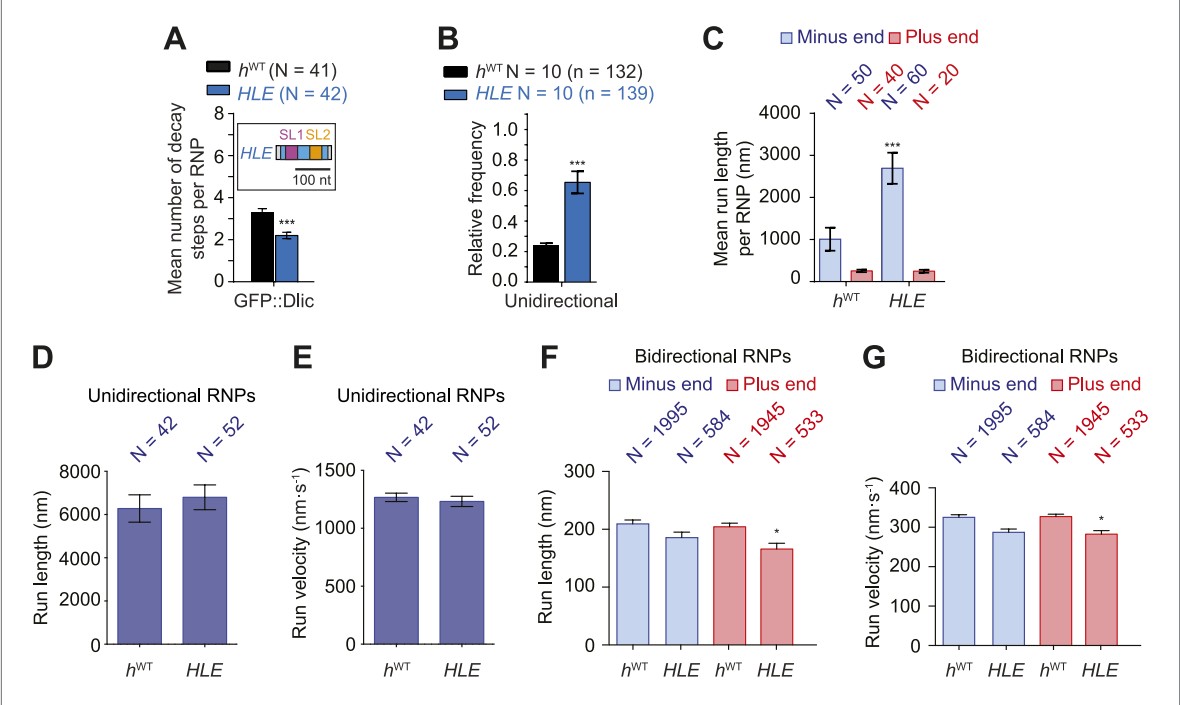

**Figure 4**. The *HLE* alone promotes unidirectional motion. (**A**) Stepwise GFP photobleaching analysis of RNPs assembled on *h*<sup>WT</sup> or *HLE* RNAs; inset, schematic of *HLE* RNA that was fused to the aptamer sequence. Note the significant decrease in the relative copy number of GFP::Dlic on the *HLE* compared to *h*<sup>WT</sup>. N, number of photobleaching traces analysed. See **Figure 4—figure supplement 1A** for distribution of values. (**B**) Mean proportion of motile *h*<sup>WT</sup> or *HLE* RNPs that are unidirectional per imaging chamber. N, number of chambers analysed; n, total number of RNPs. (**C**) Mean run length per RNP of motile *h*<sup>WT</sup> or *HLE* RNPs. N, number of RNPs analysed. (**D** and **E**) Mean length (**D**) and velocity (**E**) of individual runs of unidirectional RNPs. N, number of individual runs of RNPs (from 25 RNPs each for *h*<sup>WT</sup> and *HLE*). (**F** and **G**) Mean length (**F**) and velocity (**G**) of individual runs of bidirectional RNPs (from 40 and 20 RNPs for *h*<sup>WT</sup> and *HLE*, respectively). Images for all analyses in the figure were acquired at a lower frame rate of 4.2 fps; this is because the small number of Cy3 dyes that could be incorporated into the short *HLE* RNA necessitated imaging with a higher exposure time. Note that the measured mean run lengths of unidirectional and bidirectional RNPs of the same species are higher at 4.2 fps than at 15 fps (**Figure 3F,H**), presumably due to short pauses and frequent reversals (in the case of bidirectional RNPs) being missed at the lower frame rate. Velocity of measured runs of bidirectional RNPs of the same species is lower at 4.2 fps than at 15 fps, presumably for the same reason, whereas velocity of unidirectional runs is not significantly affected by the different frame rate as short pauses have little affect on the velocity measured for such long runs (**Figure 3G,I**). ***p<0.001; **p<0.01; *p<0.05 (Mann Whitney non-parametric *t* test), compared to *h*<sup>WT</sup> values for the same parameter; error bars represent SEM.

The following figure supplements are available for figure 4:

**Figure supplement 1**. Supplemental data on the *HLE*'s recruitment of dynein and motile properties.

similar mean lengths and velocities of individual runs in both the minus and plus end directions and these were no greater than those of the bidirectional subset of *h*<sup>WT</sup> RNPs (**Figure 4F,G**, **Figure 4—figure supplement 1E,F**). Collectively, our results suggest that the dynein-dynactin recruited specifically by RNA localisation signals determines the likelihood of RNPs initiating an exclusively unidirectional, minus end-directed mode of transport but is not sufficient to cause an increase in the length or velocity of excursions of bidirectional RNPs in either direction.

## MAPs induce reversals of *h* RNPs, with the probability of passing these obstacles not influenced by dynein-dynactin copy number

The results documented above shed light on how intrinsic properties of RNPs influence motility along the microtubule. Cargos moving along microtubules in vivo also have to contend with extrinsic factors, including MAPs that can act as obstacles. We were therefore interested in how RNPs assembled in the RAT-TRAP assay would react to the presence of MAPs and how this behaviour is influenced by manipulating dynein-dynactin copy number using the different *h* RNA variants. We decorated microtubules with two different recombinant, fluorescent MAPs and analysed the consequences on the motility of

Cy3-labelled RNPs. The first MAP was a GFP-labelled *Drosophila* kinesin-1 with a rigor mutation (Kin$_{401}$T99N:mGFP) that prevents its movement along the microtubule (*Telley et al., 2009*). The second was a human Alexa488-labelled tau (isoform 23, also known as 0N3R) (*Dixit et al., 2008*; *McVicker et al., 2011*), most of which undergoes diffusive movement on microtubules (*Hinrichs et al., 2012*) (*Figure 5—figure supplement 1A–C*). For both MAPs, stepwise photobleaching analysis of static, microtubule-associated puncta estimated an average of 4–5 monomers per diffraction-limited spot (*Figure 5—figure supplement 1D*).

The presence of kin$_{401}$T99N:mGFP on microtubules led to a concentration-dependent decrease in the net minus end directional bias of the $h^{WT}$ RNP population (*Figure 5A*). The presence of tau23 on the microtubules also reduced the overall minus end bias to $h^{WT}$ RNP motion (*Figure 5A*). This effect of the MAPs was not associated with a significant reduction in the time that $h^{WT}$ RNPs were detected in association with microtubules (*Figure 5B*). Thus, unlike the case for the wild-type kinesin-1 motor under similar conditions (*Dixit et al., 2008*; *Telley et al., 2009*; *McVicker et al., 2011*), $h^{WT}$ RNPs did not frequently detach from microtubules bound by kin$_{401}$T99N:mGFP or tau23. Thus, the presence of these MAPs appears to alter the motile properties of the RNPs without displacing them from the microtubule.

The reduction in net minus end motion of the $h^{WT}$ population in the presence of MAPs was associated with a significant decrease in the proportion of RNPs that moved exclusively unidirectionally towards the minus end of the microtubule (*Figure 5C*). Indeed, no unidirectional $h^{WT}$ RNPs were observed in chambers with the highest concentration of kin$_{401}$T99N:mGFP (*Figure 5C*). These observations suggest that unidirectional RNPs are capable of switching to a bidirectional mode in response to MAPs.

To directly observe the effects of MAPs on RNP motion we visualised individual encounters of $h^{WT}$ complexes with those regions of microtubules containing discrete puncta of kin$_{401}$T99N:mGFP or static tau23 (*Figure 5D,E*). Encounters of $h^{WT}$ RNPs with a region of microtubule bound by a kin$_{401}$T99N:mGFP punctum were rarely associated with RNP detachment or pausing (<5% of encounters in total). The frequency of such outcomes was in fact no different than expected for a segment of undecorated microtubule of similar length, as revealed by simulating the positions of MAPs on kymographs of $h^{WT}$ RNPs moving in the absence of obstacles (*Figure 5E*, *Figure 5—figure supplement 1E*). In approximately 25% of cases, RNPs passed the region bound by a rigor kinesin punctum (*Figure 5E*). Such outcomes presumably include a contribution of events in which the RNP and MAP do not meet each other due to their presence on different microtubule protofilaments. The vast majority (~70%) of encounters of $h^{WT}$ RNPs with a region bound by a kin$_{401}$T99N:mGFP punctum coincided with a reversal of the RNP on the microtubule, regardless of the direction of initial motion (*Figure 5E*). A much lower incidence of reversals (~10–15%) was expected by random chance, as revealed by analysing the dataset with simulated MAP positions (*Figure 5E*). Encounters of RNPs with regions of microtubules decorated with static tau23 also led to a reversal in the vast majority of cases, with very few detachments or pauses (*Figure 5E*). Collectively, these data demonstrate that encounters with MAPs frequently induce reversals of $h^{WT}$ RNPs.

Consistent with reversals in these experiments being influenced not just by intrinsic properties of the RNPs but also by the stochastic positioning of MAPs on the microtubule, the correlation between mean minus end and mean plus end travel distance per RNP was reduced in the presence of kin$_{401}$T99N:mGFP and tau23 (*Figure 5F*). The influence of MAPs may therefore explain why tightly correlated motion of cargo-motor complexes in each direction has not been previously observed in vivo.

We next addressed the influence of dynein-dynactin copy number on the behaviour of RNPs upon encounters with MAPs. We hypothesised that increasing dynein-dynactin copy number on RNPs could facilitate the passing of MAPs by increasing the probability of initiating motility on an obstacle-free protofilament. To test this notion, Cy3-labelled $h^{SL1x3}$ RNPs, containing significantly more dynein-dynactin copies on average than $h^{WT}$ RNPs, were observed in the presence of kin$_{401}$T99N:mGFP obstacles. Consistent with what was seen for $h^{WT}$ (*Figure 5A–C*), the proportion of $h^{SL1x3}$ RNPs that were unidirectional, and hence the overall minus end bias of the population, was reduced in the presence of the MAP without a significant reduction in the dwell time of RNPs on microtubules (*Figure 5—figure supplement 2A–C*). Thus, the motile properties of $h^{SL1x3}$ RNPs along microtubules were sensitive to the rigor kinesin.

Analysis of individual encounters of $h^{SL1x3}$ RNPs with regions of microtubules bound by kin$_{401}$T99N:mGFP puncta revealed a quantitatively similar behaviour to that observed for $h^{WT}$ RNPs;

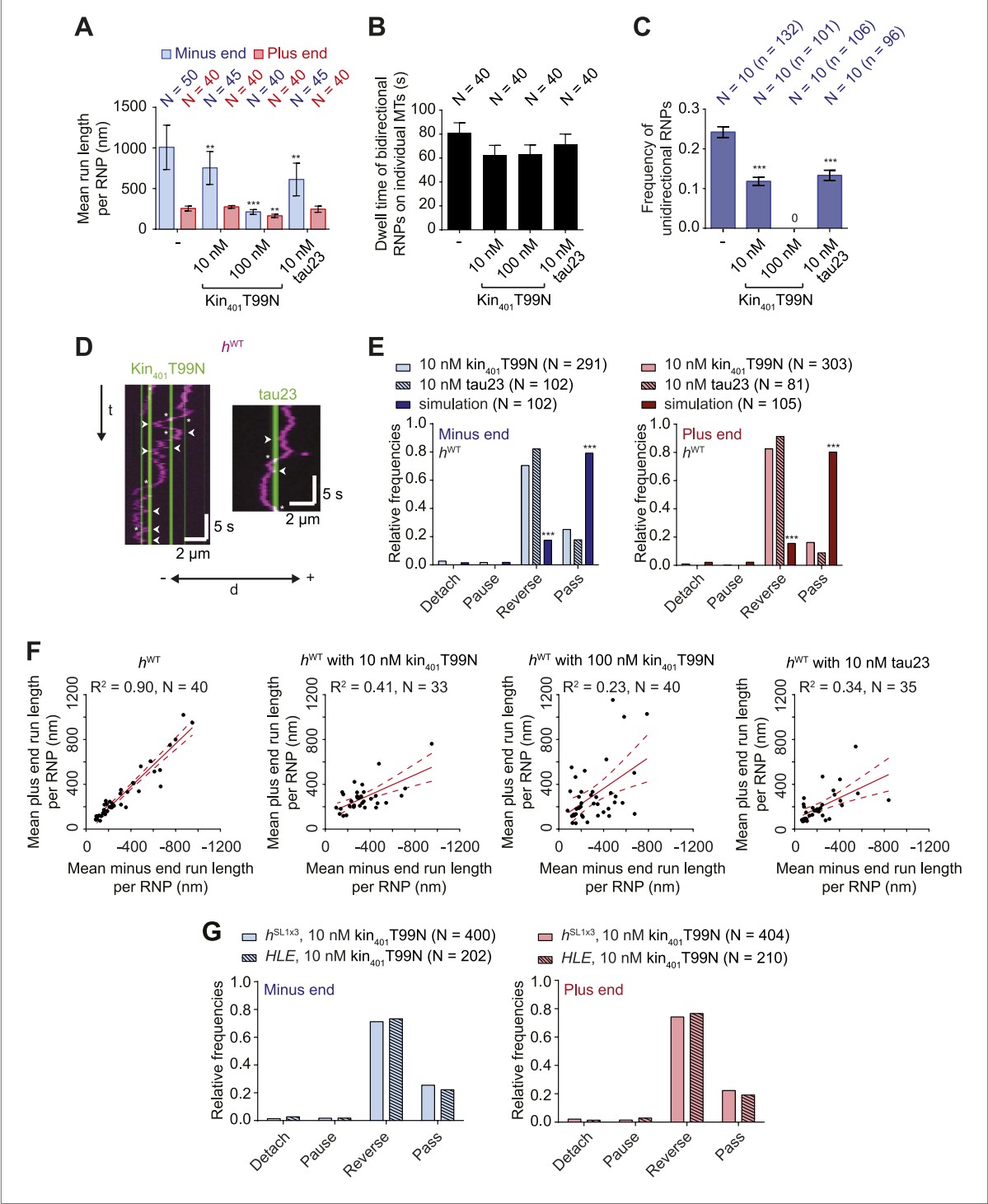

**Figure 5**. Effects of microtubule-associated proteins on the motile properties of *h* RNPs. (**A**) Mean run length per RNP for *h*[WT] RNPs when solutions of *Drosophila* GFP-tagged kinesin-1 (aa 1–401) rigor mutant (kin₄₀₁T99N:mGFP) and human Alexa488-tau23 (tau23) were previously added into the imaging chamber at the indicated concentrations. (**B**) Mean dwell time of bidirectional RNPs on individual microtubules with or without MAPs. In A and B, N is number of RNPs analysed. (**C**) Proportion of motile *h*[WT] RNPs per imaging chamber that are unidirectional in the presence and absence of MAPs. N, number of chambers analysed; n, total number of RNPs. (**D**) Kymographs exemplifying encounters of RNPs (magenta) with regions of microtubules containing kin₄₀₁T99N:mGFP or static tau23 (green). t, time; d, distance. Arrowheads and asterisks mark examples of encounters that are associated with RNP reversal or a passing event, respectively. A large fraction of tau23 underwent diffusive movement along microtubules (***Figure 5—figure supplement 1A–C***), consistent with recent observations (***Hinrichs et al., 2012***). A microtubule with a single, static patch of tau23 is therefore shown *Figure 5. Continued on next page*

*Figure 5. Continued*

here for clarity. (**E**) Outcomes of individual encounters of $h^{WT}$ RNPs with regions of microtubules associated with kin$_{401}$T99N:mGFP or the static fraction of tau23. 'Minus end' and 'Plus end' indicate the direction of RNP movement prior to the encounter. 'Simulation' refers to a dataset in which the positions of microtubule-associated kin$_{401}$T99N:mGFP puncta from an independent experiment were artificially superimposed on kymographs of $h^{WT}$ RNPs moving in the absence of added MAPs. N, total number of encounters analysed. (**F**) MAPs cause a reduction in the correlation between mean minus and plus end run length for individual bidirectional $h^{WT}$ RNPs (RNPs with ≥20 runs in total were used for this analysis). Red solid line represents the best linear fit and dashed lines represent the 95% confidence interval for the data. $R^2$, correlation coefficient; N, total number of individual RNPs analysed. (**G**) Outcomes of individual encounters of *HLE* and $h^{SL1x3}$ RNPs with regions of microtubules associated with kin$_{401}$T99N:mGFP. 'Minus end' and 'Plus end' indicate the direction of RNP movement prior to the encounter. N, total number of encounters analysed. In **A**–**C**, error bars represent SEM. \*\*\*p<0.001; \*\*p<0.01 (Mann Whitney non-parametric *t* test compared to chambers without MAPs (**A**–**C**) or Fisher's exact test compared to observed outcomes in the presence of MAPs (**E**)). In all cases, images were acquired at 4.2 fps to enable comparison with *HLE* data.

The following figure supplements are available for figure 5:

**Figure supplement 1**. Further characterisation of kin$_{401}$T99N:mGFP and tau23 puncta on individual microtubules.

**Figure supplement 2**. MAPs curtail the motility of *HLE* and $h^{SL1x3}$ RNPs on microtubules.

~70% of encounters preceded by either minus or plus end-directed motion were associated with a reversal, while detachments and pauses were rare (***Figure 5G***). The movement of RNPs containing the short *HLE* RNA was also curtailed by kin$_{401}$T99N:mGFP, with the proportion of motile RNPs that were unidirectional reduced significantly (***Figure 5—figure supplement 2A–C***). Furthermore, despite containing fewer associated dynein-dynactins, *HLE* RNPs reversed at obstacles with similar probability to $h^{WT}$ and $h^{SL1x3}$ RNPs (***Figure 5G***). Collectively, this series of experiments indicates that the total number of associated dynein-dynactin complexes does not influence the ability of RNPs to navigate around microtubule-associated obstacles.

## The behaviour of RNPs at microtubule ends

The results described above demonstrate that RNPs can respond to microtubule-associated obstacles by reversing direction. To determine whether RNPs can react to other cytoskeletal features, we studied their response to encounters with microtubule ends. Interestingly, encounters of bidirectional $h^{WT}$ RNPs with plus ends always led to a reversal in travel direction (***Figure 6A,B***). Thus, these RNPs do not detach at the plus end, a behaviour that might facilitate their long-term translocation on the microtubule cytoskeleton in vivo ('Discussion').

Reversals of bidirectional $h^{WT}$ RNPs were also common upon reaching the minus end, occurring ~70% of the time (***Figure 6A,B***). Detachments of bidirectional $h^{WT}$ RNPs upon immediately reaching a minus end were rare (~5% of total encounters) (***Figure 6B***). Approximately 25% of encounters of these RNPs with minus ends resulted in long-term pausing at this site, with a mean dwell time of ~1 min (***Figure 6B,C***). By definition, unidirectional $h^{WT}$ RNPs did not reverse at the minus end. Nonetheless, we never observed an example of a reversal at the minus end that was preceded by a long minus end-directed run (>3 µm). Thus, unidirectional motion does not appear to be followed by a reversal at the minus end. Approximately 70% of encounters of unidirectional $h^{WT}$ RNPs with a minus end resulted in pausing, with detachment from the microtubule occurring in all other cases (***Figure 6D,E***). Collectively, these data demonstrate that $h^{WT}$ RNPs are capable of recognising the minus end of microtubules and being retained there. This process may contribute to the dynein-dependent anchorage of RNAs in the apical cytoplasm that has been observed in vivo (***Delanoue and Davis, 2005***) ('Discussion').

To study the effect of dynein-dynactin copy number on the response of RNPs to microtubule ends we turned our attention to the behaviour of $h^{\Delta LE}$ and $h^{SL1x3}$ RNPs assembled in the RAT-TRAP assay. Bidirectional RNPs containing these RNA species behaved indistinguishably to $h^{WT}$ RNPs at minus and plus ends (***Figure 6B,C***). Unidirectional $h^{SL1x3}$ RNPs also responded to minus ends of microtubules in the same manner as unidirectional $h^{WT}$ RNPs (***Figure 6E,F***). Thus, average dynein-dynactin copy number does not influence the reaction to microtubule ends of RNPs that harbour *h* 3'UTRs.

Interestingly, the mean durations of pauses of unidirectional $h^{WT}$ and $h^{SL1x3}$ RNPs at minus ends were ~ fivefold shorter than those exhibited by bidirectional RNPs containing the same RNA species

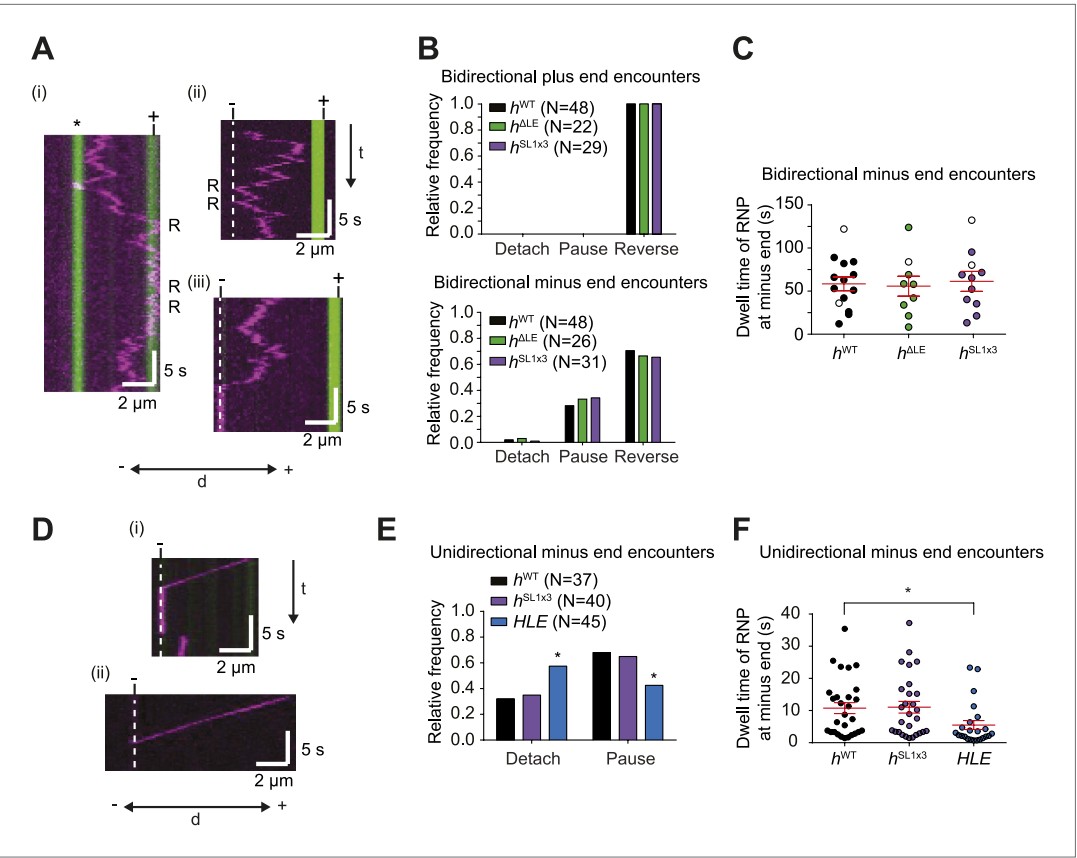

**Figure 6**. The responses of RNPs to microtubule ends. (**A**) Kymographs showing examples of encounters of bidirectional $h^{WT}$ RNPs with microtubule plus ends (i) and minus ends (ii and iii). RNPs are shown in magenta and microtubules in green (plus end labelled by greater incorporation of HiLyte 647-tubulin). '*' in (i) represents a different microtubule with plus end lying along the tracked RNP path. R indicates a number of examples of reversals. Dashed line indicates position of microtubule minus end. (**B**) Outcome of encounters of bidirectional RNPs with minus and plus ends of microtubules. Pauses were defined as events in which RNPs were stationary for longer than 1 frame (0.236 s). N, total number of encounters analysed. (**C**) Duration of minus end pausing events for different bidirectional h RNP variants. Pausing events usually ended with disappearance of the Cy3 signal (presumably due to detachment from the microtubule or Cy3 photobleaching). Open circles indicate the minority of events that ended abruptly due to completion of image acquisition. (**D**) Kymographs showing examples of encounters of unidirectional $h^{WT}$ RNPs with microtubule minus ends leading to pausing (i) or detachment (ii). RNPs, microtubules, and minus ends are depicted as in **A**. t, time; d, distance. (**E**) Quantification of outcomes of encounters of unidirectional RNPs with minus ends of microtubules (no unidirectional $h^{\Delta LE}$ RNPs were observed [**Figure 3E**]). N, total number of encounters analysed. *p<0.05 (Fisher's exact test), compared to $h^{WT}$ values for the same parameter. (**F**) Duration of pausing events for different unidirectional h RNP variants. *p<0.05 (Mann Whitney non-parametric t test), compared to $h^{WT}$ values for the same parameter. In **C** and **F**, long and short horizontal red lines demarcate the mean and SEM, respectively. Images for all of these analyses were acquired at 4.2 fps to enable comparison with HLE data.

(**Figure 6C,F**; e.g., $h^{WT}$ unidirectional, 10.7 ± 1.7 s; $h^{WT}$ bidirectional, 58.9 ± 8.0 s (p<0.0001, Mann–Whitney non-parametric t test)). This finding led us to consider the possibility that features that promote bidirectional motion aid in the retention of RNPs at the minus end. To explore this hypothesis further, we examined the behaviour at minus ends of the HLE element, which lacks sequences within the rest of the 3'UTR that can recruit dynein-dynactin and drive bidirectional motion (as revealed by analysis of the $h^{\Delta LE}$ RNA [**Figure 3**]). Upon reaching the minus end, unidirectional HLE RNPs were more likely to detach than unidirectional $h^{WT}$ and $h^{SL1x3}$ RNPs (**Figure 6E**), and the duration of pauses was significantly reduced (**Figure 6F**). This observation is compatible with the notion that anchorage of RNPs at minus ends is facilitated by features that promote bidirectional movement.

## Discussion

### Insights into the mechanisms of cargo sorting by multi-motor assemblies

Several previous studies have manipulated the copy numbers of cargo-associated motors in order to elucidate how multiple motors orchestrate sorting. One experimental approach has been to alter the numbers of isolated motors or motor domains attached to artificial cargos such as beads or DNA origami (e.g.,*Block et al., 1990*; *Mallik et al., 2005*; *Diehl et al., 2006*; *Vershinin et al., 2007*; *Derr et al., 2012*; *Furuta et al., 2013*). Although very informative, these studies did not include physiological cargo, cargo adaptors, and motor co-factors that could potentially modulate motor behaviour. Other studies have used genetic manipulations to alter the numbers of motor complexes available to cargos in vivo (*Shubeita et al., 2008*; *Reis et al., 2012*). These approaches are physiologically relevant but cannot rule out influences from the cellular environment, including possible indirect effects of altered motor concentration on other processes that impinge on cargo motility. In this study, we have manipulated the copy number of native motor complexes on a physiological type of cargo, by incubating cellular extracts with RNA variants, and studied the consequences on motility in a defined in vitro setting using high spatiotemporal resolution imaging.

We find that *h* wild-type RNPs associate with dynein-dynactin and can undergo either unidirectional motion in the minus end direction that is highly processive or bidirectional motion that has characteristics of a diffusive process. Our experiments indicate that unidirectional RNP movement is driven by active, ATP hydrolysis-mediated translocation of dynein along the microtubule. The most parsimonious explanation for the bidirectional motion is that it is also due to dynein undergoing back-and-forth movements along the microtubule, a behaviour that has been observed in several studies of the purified motor in vitro (e.g., *Wang et al., 1995*; *Wang and Sheetz, 1999*, *2000*; *Mallik et al., 2005*; *Ross et al., 2006*; *Miura et al., 2010*) and also appears to occur in vivo (*Ananthanarayanan et al., 2013*). Indeed, we observed bidirectional movement of a significant subset of RNPs containing only the *HLE* (~30% in the absence of MAPs and ~60% in their presence [*Figure 5—figure supplement 2C*]), on which we failed to detect binding of a kinesin family member under conditions in which dynein and dynactin were readily detected (*Dix et al., 2013*). Interestingly, bidirectional motion of RNPs is not overtly sensitive to inhibition of dynein's ability to hydrolyse ATP. Thus, our findings are compatible with those of *Miura et al. (2010)* who reported passive diffusion along microtubules of dynein in complex with a dynactin component in the presence of ATP or ATP-vanadate. We found no evidence that bidirectional RNPs can undergo long, ATP hydrolysis-dependent runs in both directions akin to those documented for individual, GFP-labelled dynein-dynactin complexes purified from mouse brain (*Ross et al., 2006*).

Varying the number of SL1 elements within the context of the *h* 3′UTR revealed a correlation between the total copy number of dyneins per RNP and the probability of entering into the unidirectional, minus end-directed state. Previous studies using purified dynein bound to artificial cargos have demonstrated that increasing motor copy number is sufficient to stimulate processive movement towards minus ends (*Mallik et al., 2005*; *Ross et al., 2006*). However, experiments with the isolated *HLE* signal indicated that total motor number is not the key determinant of the unidirectional mode of RNP movement. The *HLE* alone has a statistically indistinguishable copy number of dynein components to the *h*[ΔLE] RNA, in which the localisation signal has been replaced by a heterologous sequence in the context of the *h* 3′UTR, yet only the former RNA is capable of unidirectional motion. These data suggest that features associated with the RNA signal are sufficient to increase the probability of processive movement of the associated dynein. It has recently been shown in *Schizosaccharomyces pombe* that microtubule-associated dynein can switch from diffusive to processive behaviour upon contacting cortical anchors, an event that regulates the generation of pulling forces on the microtubule (*Ananthanarayanan et al., 2013*). Our data suggest that regulation of dynein processivity by associated factors may be a widespread phenomenon.

Collectively, our results support a novel model in which the same cargo species can interact with processive or non-processive dynein, with discrete cargo-associated features regulating the probability of switching between the two behaviours (*Figure 7*). Thus, the regulatory mechanisms underpinning sorting of RNPs in this system appear distinct from those of other well-studied bidirectional cargos, which involve the interplay of opposite polarity force-generating motors, such as dyneins and kinesins (*Shubeita et al., 2008*; *Ally et al., 2009*; *Soppina et al., 2009*; *Jolly and Gelfand, 2011*; *Reis et al., 2012*).

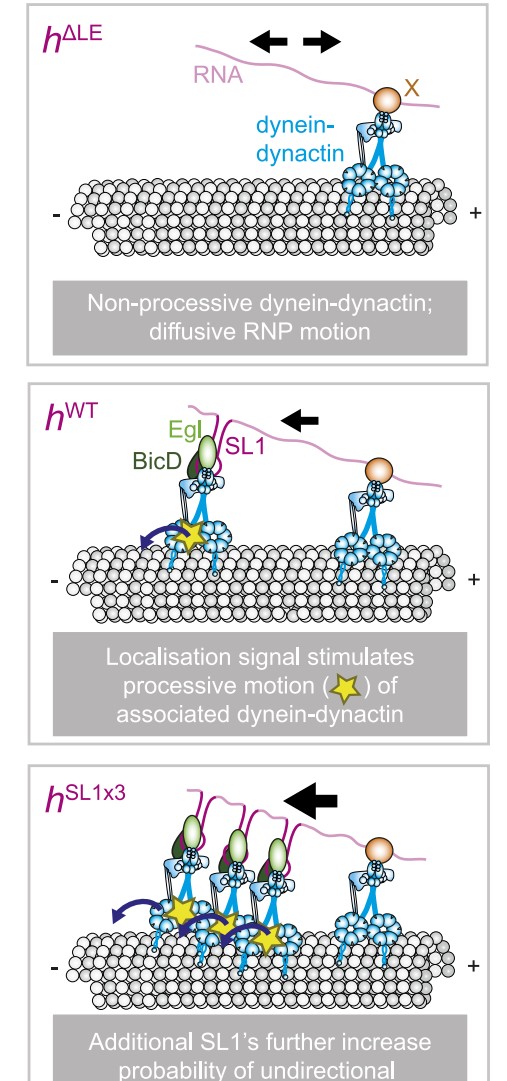

**Figure 7**. Schematic model for the role of RNA localisation signals in controlling net RNP motion. (Top) In the absence of RNA localisation signals, RNPs undergo passive diffusion on the microtubule lattice. Non-processive dyneins are recruited to sites in the RNA other than localisation signals through an unknown factor (X). The most parsimonious explanation for the diffusive movement is that it is driven by dynein-dynactin. (Middle) The presence of a localisation signal leads to the recruitment of additional dynein-dynactin and a subset of RNPs in the population undergoing processive, minus end-directed transport. This introduces a minus end bias to motion of the RNA population. It is not the increased total copy number of the motor complex that drives unidirectional movement, but rather the ability of localisation signals to increase the probability of associated dynein-dynactin

*Figure 7. Continued on next page*

How might localisation signals regulate dynein processivity? One possibility is that proteins recruited by the localisation signal directly regulate the activity of the motor. Two candidates to serve such a role are the adaptor proteins Egl and BicD, which are associated only with the dynein-dynactin bound to the localisation signals (*Bullock et al., 2006*; *Dix et al., 2013*). It is also conceivable that the structure or rigidity of the RNA signal plays an architectural role in presenting dynein-dynactin to the microtubule in a manner that favours processive movement.

## Insights into how cargo-motor complexes navigate the cytoskeletal network

Our study also sheds light on how physiological cargo-motor complexes respond to extrinsic factors within the cytoskeletal environment. We show that RNPs frequently reverse when encountering the regions of microtubules bound by puncta of both MAPs studied, rigor kinesin and tau23. The behaviour of RNPs at MAPs is therefore highly reminiscent of that seen for individual, purified dynein-dynactin complexes in vitro (*Dixit et al., 2008*). Interestingly, comparison of the behaviour of the *h* RNA variants reveals that the number of dynein-dynactins associated with an RNP does not increase the probability of passing microtubule-associated obstacles.

The ability of RNPs to reverse at MAPs may help them navigate to their destination in vivo. For example, reversals upon meeting an obstacle may facilitate encounters of RNPs with intersecting microtubules. Switching of RNPs between microtubules, a behaviour we have observed when intersections occur in our in vitro assays (unpublished observations), could allow these complexes to explore alternative routes to their destination. Reversals of RNPs at a MAP may also give dynein space to switch to a different lateral position on the same microtubule and thereby provide another opportunity to pass the obstacle following resumption of movement in the previous direction. Compatible with this notion, we see RNPs moving on single microtubules that can pass obstacles after multiple attempts (e.g., *Figure 5D*).

Our analysis of the behaviour of RNPs at microtubule ends indicates that plus end encounters always result in a reversal. The behaviour may also be advantageous in a cellular environment by preventing detachment of RNPs at this point, an outcome that would necessitate a rebinding event before motion on the microtubule network can resume. Interestingly, a subset of RNPs undergoes pausing at the minus ends of microtubules,

*Figure 7. Continued*

entering into a processive state. (Bottom) Addition of more SL1 elements increases further the likelihood of RNPs entering into the unidirectional state, leading to a greater minus end bias to motility of the RNA population. It is not known whether unidirectional motion involves stepping of a single dynein-dynactin or of multiple motor complexes. Resolving this issue will require long-term nanoscale analysis. Note that for simplicity the cartoon does not attempt to depict the absolute copy number of dynein-dynactin complexes on each RNP.

with a mean dwell time of ~1 min for bidirectional complexes. These findings demonstrate that the probability of changing directions is different at the minus end and the plus end of the microtubule and that additional in vivo features, such as the γ-tubulin ring complex or other centrosome-associated factors, are not obligatory for long-term retention of RNPs at the minus end. Intrinsic behaviours of RNPs upon reaching the minus end of the α/β-tubulin polymer may therefore contribute to the dynein-dependent anchorage of RNAs in the vicinity of minus ends in vivo (*Delanoue and Davis, 2005*). Analysis of $h^{ALE}$, $h^{WT}$, and $h^{SL1x3}$ RNAs demonstrates that the probability of an RNP undergoing minus end pausing in vitro, as well as the duration of such events, is not influenced by the addition of more dynein-dynactins through localisation elements. This finding offers an explanation for why inhibition of Egl and BicD following translocation of localising RNAs to the apical cytoplasm of the embryo does not affect dynein-dependent anchorage (*Delanoue and Davis, 2005*).

It is intriguing that the average dwell time of pausing events of unidirectional $h^{WT}$ and $h^{SL1x3}$ RNPs at minus ends is ~ fivefold less than that of bidirectional RNPs harbouring the same RNA species. We also find that the unidirectional *HLE* RNPs, which lack features within the *h* 3'UTR that can recruit non-processive dynein-dynactin, dwell at the minus end for significantly less time than the unidirectional $h^{WT}$ and $h^{SL1x3}$ RNPs. One explanation for these findings is that the ability of dynein bound to localisation signals to walk processively off the minus end is antagonised by interactions with the microtubule mediated by non-processive dynein bound at other sites in the RNA. Additional, long-term experiments will be required to test this hypothesis. Nonetheless, our data suggest more generally that features that promote bidirectional motion could assist in the retention of RNPs at minus ends.

Collectively, our analysis of encounters of RNPs with MAPs and microtubule ends raises the possibility that the co-existence of unidirectional or bidirectional modes of movement facilitates efficient navigation of an RNA population to its destination in vivo. Processive, unidirectional movement in the minus end direction could be beneficial for rapid, directional movement along regions of the microtubule that are not rich in MAPs. Diffusive motion along microtubules may be valuable for movement of RNPs through an obstacle rich environment and could still contribute to asymmetric sorting as it is associated with long-term retention of complexes at microtubule minus ends. This strategy appears analogous to that used by DNA enzymes and kinesins that depolymerise microtubules, which can employ one-dimensional diffusion to search for their specific target sites (*Helenius et al., 2006*; *Gorman and Greene, 2008*).

## Perspective

Our in vitro work on RNA motility has provided several new insights into how cargo-motor complexes operate and how their behaviour is modulated by encounters with the environment. Our data lead to a model in which discrete cargo-associated features regulate motor processivity, a phenomenon that could not have been recapitulated using minimal motor elements coupled to artificial cargos. Our results also illustrate that intrinsic motile properties of cargo populations in vivo are likely to be obscured by the influence of extrinsic factors including MAPs and microtubule ends. Further exploitation of the RAT-TRAP assay is likely to be an effective strategy for shedding light on molecular mechanisms that underpin intrinsic and extrinsic regulation of cargo motility, particularly when combined with powerful *Drosophila* gene perturbation techniques. In the longer term, it will be important to understand how the behaviours defined in vitro are integrated during sorting of single RNA molecules in vivo, a goal that necessitates the development of new methods to visualise movement of transcripts in the optically challenging embryo system.

## Materials and methods

### *Drosophila* strains

Wild-type embryos were of the OR-R strain. *P(Ubi-GFP::Dlic)* (*Pandey et al., 2007*) and *P(mat-tub-α4-GFP::Dmn)* (*Januschke et al., 2002*) lead to ubiquitous expression of the GFP fusion proteins during

early development and were gifts from J Raff (Oxford University, UK) and A Guichet (Institut Jacques Monod, France), respectively.

## Fluorescent RNA synthesis

The *hairy* RNA ($h^{WT}$) used for in vitro motility assays is 822 nt long and represents the majority of the transcript's 3'UTR. $h^{SL1x3}$ is a *h* 3'UTR variant in which the 124 nt *h* localisation signal (*HLE*) is replaced by a cassette containing three copies of the 46 nt *h* stem-loop 1 (SL1; *Bullock et al., 2003*) separated by 12–17 nt single stranded spacers. The presence of three SL1 elements increases the net minus end bias to *h* 3'UTR RNP motility in vivo compared to the wild-type *HLE* (*Bullock et al., 2006*). $h^{\Delta LE}$ is a *h* 3'UTR variant in which the *HLE* is replaced with the same-sized piece of RNA derived from the glutathione-S-transferase (GST) gene from *Schistosoma japonicum* (generated by PCR from the pGEX6P-1 vector [GE Healthcare]). *HLE* has the isolated 124 nt *h* localisation signal, together with 25 nt of the 3' sequence of the *h* 3'UTR (the inclusion of the additional sequence was designed to assist folding of the SL2 element of the *HLE*). Sequences encoding $h^{WT}$, $h^{SL1x3}$, $h^{\Delta LE}$, and *HLE* RNAs were introduced into the pTRAPv3.0 vector (Cytostore), allowing a fusion RNA to be synthesised from the T7 promoter that contains two copies of the S1 streptavidin-binding aptamer (*Srisawat and Engelke, 2001*) 5' to the RNA of interest (the transcription product from the T7 promoter until the start of *h* RNA variants is 264 nt long). We also introduced 15 nt of RNA sequence at the 5' and 3' of the *aptamer-HLE* (GCATACCGGATACGC and CCATAGGCATAGCGC, respectively) as part of the splint ligation procedure for end-labelling the RNA with Cy3 dyes (see below).

Cy3-labelled, aptamer-linked $h^{WT}$, $h^{SL1x3}$, and $h^{\Delta LE}$ RNAs were synthesised with the MEGAscript T7 kit (Ambion, Foster City, CA) as per the manufacturer's instructions with 1.875 mM Cy3–UTP (PerkinElmer, Waltham, MA) and 5.625 mM unlabelled-UTP (Roche, Switzerland). Alexa488-labelled, aptamer-linked $h^{WT}$ RNA was synthesised as described previously (*Bullock et al., 2006*). Unincorporated nucleotides were removed using mini Quick Spin RNA Columns (Roche). The degree of body labelling of RNAs with these dyes was determined with a Nanodrop spectrophotometer (Thermo Scientific, Waltham, MA). Typically, there was an average of ~8 dyes per *aptamer-h* 3'UTR RNA molecule. *Aptamer-HLE* RNA was labelled at each end with Cy3 using the splint ligation technique, as previously described (*Moore and Query, 2000*). Briefly, this method uses bridging DNA oligonucleotides to facilitate T4 DNA ligase-mediated ligation of Cy3-labelled 15 nt RNA oligonucleotides (Integrated DNA Technologies, Coralville, IO) to the 5' and 3' ends of the *aptamer-HLE* RNA, followed by the removal of the DNA using DNaseI (Agilent, Santa Clara, CA).

## Flow chamber preparation and TIR microscopy imaging

Biotin-PEG-functionalised glass and passivated counter glass surfaces were prepared as described (*Bieling et al., 2010*; *Roostalu et al., 2011*). Imaging was performed at 24 ± 1°C with a total internal reflection fluorescence microscope (Nikon, Netherlands) equipped with a 100× objective (Nikon, 1.49 NA Oil, APO TIRF), using the following lasers: 150 mW 488 nm, 150 mW 561 nm laser (both Coherent (Santa Clara, CA) Sapphire), and 100 mW 641 nm (Coherent Cube). Images were acquired with a back illuminated EMCCD camera (iXon$^{EM}$+ DU-897E, Andor, UK) controlled with μManager software (http://micro-manager.org/wiki/Micro-Manager). The size of each pixel was 105 × 105 nm.

## Microtubule polymerisation and adsorption to glass

Porcine tubulins and polymerisation buffers were purchased from Cytoskeleton, Inc. (Denver, CO). Biotinylated, GmpCpp-stabilised microtubules with plus ends marked by greater incorporation of HiLyte 647 were polymerised as previously described (*Roostalu et al., 2011*). Briefly, long, biotinylated microtubules that were dimly labelled with fluorophore were first produced by polymerisation for 2 hr at 37°C from 1.66 μM unlabelled tubulin, 0.4 μM biotinylated tubulin, 0.15 μM HiLyte 647 tubulin in the presence of 0.5 mM GmpCpp (Jena Bioscience, Germany) in BRB80 (80 mM PIPES, 4 mM MgCl$_2$, 1 mM EGTA, pH 6.8). Polymerised microtubules were sedimented by centrifugation at 20,800×*g* for 8 min at room temperature. In a second step, a short microtubule plus end segment was generated that was brightly labelled with fluorophore. This was achieved by incubation of the dimly labelled microtubules for 30–45 min at 37°C in 'bright elongation mix' consisting of 1.5 μM NEM-tubulin (n-ethylmale-imide-tubulin prepared as previous described (*Phelps and Walker, 2000*)), 1.33 μM HiLyte 647 tubulin and 0.5 mM Gmp-Cpp in BRB80. Polarity-marked microtubules were then sedimented at 20,800×*g* for 8 min at RT and resuspended in BRB80 containing 40 μM taxol. Microtubules were immobilised on

a biotin-PEG-coated glass surface via streptavidin as described (*Bieling et al., 2010*). In control experiments, microtubule length was determined to be 15 ± 1.6 μm (mean ± SEM, N = 30).

## In vitro RNA transport after tethered RNA purification (RAT-TRAP)

50 μl of M-280 streptavidin-coupled Dynabeads (Invitrogen, Carlsbad, CA) were washed twice for 10 min in 1 mg·ml$^{-1}$ bovine serum albumin (BSA) on a roller mixer at room temperature. Magnetic beads were then incubated with 1 mg·ml$^{-1}$ BSA on ice for 60 min. This procedure was designed to block unspecific binding sites on the beads. Subsequently, 1 pmol of fluorescently labelled RNA (containing the streptavidin aptamers and RNA of interest) in DXB buffer (30 mM HEPES at pH 7.3, 50 mM KCl, 2.5 mM MgCl$_2$, 250 mM sucrose, 5 mM dithiothreitol, 10 μM MgATP, and Complete [EDTA-free] protease inhibitor [Roche]) with 40 U RNasin RNase inhibitor (Promega, Madison, WI) was incubated with the blocked Dynabeads for 2 hr at 4°C. *Drosophila* embryo extracts were produced from dechorionated 0–6 hr wild-type embryos by homogenising in DXB buffer as described (*Bullock et al., 2006*) using 100 μl of DXB buffer per 50 mg of embryos. Embryo extract was then centrifuged for 8 min at 9000 rpm at 4°C to remove debris. Typically, 100 μl embryo extract was added to the Dynabead–RNA complex, supplemented with 40 U of RNasin and incubated for 1 hr at 4°C on a roller mixer. The magnetic beads were washed once with DXB buffer at 4°C and RNA-motor complexes eluted by incubating the Dynabeads with 200 μl of 10 mM biotin (Invitrogen) in motility buffer (30 mM HEPES/KOH, 5 mM MgSO$_4$, 1 mM DTT, 1 mM EGTA, 0.5 mg/ml bovine serum albumin, pH 7.0) at 15°C for 20 min in a Thermomixer comfort (Eppendorf, UK; 950 rpm). Biotin competes for the interaction of the aptamer with streptavidin. In control experiments, we confirmed by immunoblotting that components of RNA-motor complexes are present in the eluate from $h^{WT}$ RNA affinity purification experiments by probing for Egalitarian, Bicaudal-D and Dynein heavy chain. These factors were not present when RNA was omitted from the procedure. For motility assays the eluate was transferred immediately to ice and subsequently introduced along with 2.5 mM ATP and an oxygen scavenging system (1250 nM glucose oxidase, 140 nM of catalase, 71 mM 2-mercaptoethanol, and 24.9 mM glucose) into a flow chamber with polarity-marked, HiLyte 647-labelled microtubules pre-adsorbed to the coverslip. In a subset of experiments sodium orthovanadate or apyrase was added to the eluate (to a final concentration of 100 μM or 20 U·ml$^{-1}$, respectively) before addition to the flow cell. In the apyrase experiment the eluate was not supplemented with ATP. Microtubules and Cy3-labelled RNA molecules were visualised with a TIR microscope (15 fps, that is 66 ms capture time (63 ms exposure plus 3 ms image acquisition) or 4.2 fps, that is 236 ms capture time (200 ms exposure plus 36 ms image acquisition)); unless stated otherwise, ~ 1500 frames or ~700 frames were collected for 15 fps or 4.2 fps imaging, respectively. *HLE* RNPs were analysed at the lower frame rate (4.2 fps). This was because the reduced number of Cy3 dye molecules per RNA molecule compared to *h* 3'UTR RNAs (see above) necessitated a longer exposure time. Experiments in which the motility of *h* 3'UTR species was compared to that of *HLE* were also performed at 4.2 fps. Hence, all motility experiments in *Figures 4–6* (involving *HLE*) were performed at 4.2 fps, with all other experiments performed at 15 fps.

## Analysis of RNP motion

For each condition/RNA species, data were collected from three independent days of experiments (8–10 imaging chambers). Random microtubules in the region-of-interest were selected, followed by analysing the RNPs that moved along them. The movement of RNPs with reference to the polarity of microtubules were analysed using TrackMate in Fiji (http://fiji.sc/TrackMate) and a custom Matlab code (available on request). Sub-pixel XY coordinates of motile RNPs were acquired from TrackMate as described (http://fiji.sc/TrackMate) using two-dimensional Gaussian fitting. The RNP coordinates were imported into Matlab where they were projected onto a vector along the coordinates of the microtubule (*Hendricks et al., 2010*; *Rai et al., 2013*), thereby producing on-axis positions of the RNP with respect to the track. The difference in on-axis position between consecutive frames (termed *d*) was then calculated. The tracking accuracy was determined to be 11.3 nm (i.e., the standard deviation of instantaneous displacements of tracked immobile RNPs on microtubules [*Hendricks et al., 2010*; *Figure 1—figure supplement 1C,D*]). Runs were defined as the sum of consecutive displacements >22 nm in one direction (i.e., before termination by pausing, reversal, RNP detachment, or the end of imaging). 22 nm was chosen as a cut-off as it is approximately twice the standard deviation of instantaneous displacement measurements for immobile RNPs (*Figure 1—figure supplement 1D*).

A similar calculation was previously used to produce a cut-off for the analysis of the motility of neuronal vesicles in vitro (*Hendricks et al., 2010*). Pauses were defined as ≥1 frame with an instantaneous displacement ≤22 nm. Reversals within RNP tracks were defined as instances when $d$ of consecutive frames $i$ and $i+1$ were >22 nm and the absolute value of the sign of $d_i$ minus $d_{i+1}$ equalled 2 (i.e., $|\text{sign}(d_i)\text{-sign}(d_{i+1})| = 2$), where $\text{sign}(d) = +1$ for $d > 0$ and $\text{sign}(d) = -1$ for $d < 0$. The trajectories were then analysed for run length and persistence time and these values used to calculate run velocity. In control experiments, analysis of 20 s bins of RNP motion over the duration of image acquisition (100–120 s) revealed that mean run length and velocity does not decrease over time. This indicates that RNA-motor complexes are relatively stable over the total period of image acquisition.

The proportion of motile RNPs that were unidirectional vs bidirectional for each RNA variant or condition was determined by manual analysis of 6–10 imaging chambers from three independent days of experiments. Automatic tracking of a subset of RNPs defined as unidirectional by manual analysis confirmed that they do not contain plus end-directed events. For the analysis in *Figure 2G*, RNPs were deemed as unidirectional, bidirectional, or stationary (no motion beyond 1 pixel) by manual analysis of kymographs. 12 randomly selected microtubules from at least three different imaging chambers were analysed for each condition.

Mean squared displacement (MSD) analysis was performed with sub-pixel resolution using randomly selected bidirectional and unidirectional RNPs filmed at 15 fps. MSD was calculated in Matlab using internal averaging (averaging over all pairs), to ensure each data point was weighted evenly, for no greater than one quarter of the total travel time, as previously described (*Saxton, 1997*). Diffusion coefficients (D) of bidirectional RNPs were calculated from a linear fit to the MSD(t) data using the equation MSD = 2Dt.

The plots in *Figures 2B–E and 5F* were produced by obtaining the mean of the individual minus and plus end run lengths or velocities for each RNP that had a total number of runs ≥20.

For the experiments documented in *Figure 1—figure supplement 1A,B*, 1 pmol each of Alexa-488-labelled *h* and Cy3-labelled *h* were mixed and captured on streptavidin beads before incubation with embryo extract. The assembled RNA-motor complexes were eluted using biotin and injected into a flow cell containing HiLyte 647-labelled microtubules as describe above. Images of microtubules were captured at the beginning and end of a series of alternating images of the Alexa-488 and Cy3 signals (236 ms capture [200 ms exposure +36 ms image acquisition] per channel, 500 frames).

## Evaluating the effects of MAPs and microtubule ends on RNP motility

DNA coding for the first 401 amino acids of conventional kinesin (kinesin-1) from *Drosophila melanogaster* with monomeric GFP fused to the C-terminal end and His6-z- to the N-terminal end was a gift from T Surrey (*Telley et al., 2009*). We used Quikchange mutagenesis (Stratagene) to introduce a T99N substitution in the kinesin motor domain to obtain a non-motile rigor mutant that constitutively associates with the microtubule in a strongly bound state (*Telley et al., 2009*). The recombinant fusion protein was expressed in bacteria (BL21(DE3) strain) and the dimeric fraction purified using gel filtration as described previously (*Telley et al., 2009*). Human tau23 (352 AA, 0 inserts and 3 repeats–0N3R, shortest isoform) was expressed in bacteria (BL21(DE3) strain), purified, and fluorescently labelled with Alexa-488 maleimide as previously described (*Goedert and Jakes, 1990*; *McVicker et al., 2011*). Tau23 contains one reactive cysteine, at position 322 within the 3rd repeat. The percentage of protein that was labelled, determined using a NanoDrop ND-1000 spectrophotometer (Thermo Scientific), was ~80%.

Kin$_{401}$T99N:mGFP was introduced into chambers containing HiLyte 647-microtubules preadsorbed to the coverslip. After washing with 5x chamber volume of motility buffer, RNA-motor complexes were introduced with ATP and oxygen scavenging agent as described above. A still image of the microtubule (635 nm excitation, 236 ms capture time [200 ms exposure +36 ms image acquisition]) was acquired, followed by a still image of rigor kin$_{401}$T99N:mGFP (488 nm excitation, 1.036 s capture time [1 s exposure +36 ms image acquisition]). This was followed by continuous imaging of the Cy3-RNA channel (561 nm excitation, 236 ms capture time [200 ms exposure +36 ms image acquisition]) for 700 frames to monitor the behaviour of RNA-motor complexes. At the end of this series another still image of kin$_{401}$T99N:mGFP was acquired as above. Encounter statistics in *Figure 5E,G* were calculated manually using kin$_{401}$T99N:mGFP present at the same position during the entire time interval of the movie. We had previously confirmed using continuous imaging of the GFP channel that kin$_{401}$T99N:mGFP

rarely moved during several minutes of filming. Outcomes of encounters were scored when the RNP reached the pixels containing $kin_{401}$T99N:mGFP fluorescent signal on the kymograph. The same experimental set up was used for tau23 experiments, with the following modifications: a movie was recorded (488 nm excitation, 236 ms capture time [200 ms exposure +36 ms image acquisition], 40 frames) before and after imaging the RNA-motor complexes. This allowed assessment of which tau23 patches were stationary and only these were used for encounter analysis. Analysis of run lengths and velocities of RNPs in the absence and presence of MAPs was performed using automated tracking and analysis as described above.

To quantify simulated encounters of RNPs with $kin_{401}$T99N:mGFP (*Figure 5E*, *Figure 5—figure supplement 1E*), kymographs of static, microtubule-associated $kin_{401}$T99N:mGFP puncta generated in independent experiments were superimposed on kymographs of RNPs moving on microtubules without any MAPs added to the chamber. Scoring of encounters of RNPs with the simulated obstacles was performed as described above.

Outcomes of RNP encounters with microtubule ends were scored manually from kymographs. Pauses were defined as events in which RNPs were stationary for longer than 1 frame (0.236 s).

## Stepwise photobleaching

Relative copy numbers of GFP-tagged Dlic and Dmn on RNA-motor complexes were estimated using RNPs assembled and eluted using the same protocol as for RAT-TRAP assays, except no ATP was added to the eluate before introduction into the imaging chamber. The absence of nucleotide was designed to promote stable binding of RNA-motor complexes to microtubules. Note that quantification of the number of GFP molecules in RNPs undergoing motion was precluded by fluctuations in GFP fluorescence intensity. GFP and Cy3 fluorophores were illuminated sequentially for 136 ms and images captured as described above. Positions of HiLyte 647-labelled microtubules were recorded by image capture at the beginning and end of filming. Kalaimoscope Motion Tracker (Transinsight, Germany) was used to plot the change over time in fluorescence intensities of GFP signals co-localised with Cy3-RNA signal on microtubules (after background subtraction). The number of photobleaching steps (discrete decay steps of fluorescent signal intensity) in each trace that reached baseline levels of fluorescence was determined manually, following published procedures, after a Chung-Kennedy filter was applied on the traces as described (*Chung and Kennedy, 1991*; *Ulbrich and Isacoff, 2008*; *Jain et al., 2011*; *Badrinarayanan et al., 2012*). We confirmed the accuracy of manual scoring by analysing a subset of the traces using the step-fitting algorithm StepFinder (*Kerssemakers et al., 2006*). Fluorescence traces showing a sudden, large drop in intensity to basal levels were not used. These events, which happened very rarely, could indicate dissociation of the RNA-motor complex from the microtubule. Traces showing excessive fluorescence fluctuation were also excluded from analysis. Background fluorescence values were determined by averaging the fluorescence value from 8–10 randomly chosen pixels in the field-of-view.

To estimate the copy numbers of $kin_{401}$T99N:mGFP or tau23 per diffraction-limited spot, proteins were introduced into an imaging chamber (in the absence of ATP) containing HiLyte 647-labelled microtubules bound to the glass, together with oxygen scavenging agent (at the same concentration as above). Images were captured (488 nm excitation, 136 ms capture [100 ms exposure +36 ms image acquisition]) until the fluorescent signal reached basal levels (indicating photobleaching or detachment). Estimation of the number of GFP photobleaching steps was performed manually as describe above, with the same criteria used for exclusion of a subset of traces from the analysis.

## Immunoblotting

To evaluate the ratio of GFP-labelled to unlabelled Dlic in extracts of GFP::Dlic embryos we performed immunoblotting with a mouse anti-Dlic P5F5 antibody (*Mische et al., 2008*) (a gift from T Hays, Minnesota University, USA; used at a 1:1500 dilution). Signal was detected, used Alexa488-conjugated anti-mouse secondary antibodies (1:1000; Invitrogen), and a Typhoon imaging system (GE Healthcare, UK). Quantification of the ratio of unlabelled Dlic to GFP-labelled Dlic was performed using ImageJ (http://rsb.info.nih.gov/ij/) following background subtraction.

## Statistics

Data plotting and curve fitting was performed with GraphPad (La Jolla, CA) Prism 6 and Matlab R2012b (Mathworks, Natick, MA). Evaluations of statistical significance are described in the appropriate legend.

## Acknowledgements

We are very grateful to M Amrute-Nayak, A Carter, C Dix, C Duellberg, R Hegde, A Hendricks, Y Liu, S McLaughlin, and KV Parag for technical advice, G Fraser, A Guichet, T Hays, J Raff, and T Surrey for reagents, J Kerssemakers for providing the Stepfinder algorithm and HT Hoang and R Hegde for comments on the manuscript. This work was supported by UK MRC core funding (U105178790), a Lister Institute Research Prize (both to SLB), a LMB Cambridge Scholarship and a Cambridge Commonwealth and Cambridge Overseas Trust Scholarship (both to HCS).

## Additional information

### Funding

| Funder | Grant reference number | Author |
|---|---|---|
| Medical Research Council | U105178790 | Harish Chandra Soundararajan, Simon L Bullock |
| Lister Institute of Preventive Medicine | | Simon L Bullock |

The funders had no role in study design, data collection and interpretation, or the decision to submit the work for publication.

### Author contributions

HCS, Conception and design, Acquisition of data, Analysis and interpretation of data, Drafting or revising the article; SLB, Conception and design, Analysis and interpretation of data, Drafting or revising the article

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
