## [Decision Letter]

Thank you for sending your work entitled ”Intrinsic and Extrinsic Control of Dynein-Based Cargo Transport Revealed by a Novel In Vitro Assay for mRNP Motility” for consideration at *eLife*. Your article has been favorably evaluated by the Editor-in-Chief Randy Schekman and 3 peer reviewers.

The Editor-in-Chief and the reviewers discussed their comments before we reached this decision, and the Editor-in-Chief has assembled the following comments to help you prepare a revised submission.

Understanding how microtubule-based cargos move bidirectionally along their tracks and achieve distinct patterns of spatial organization is an important problem. Here, Soundararajan and Bullock tackle this problem by first developing an improved in vitro motility assay for analyzing microtubule-based RNP motility. Their method relies on purifying native motor complexes from Drosophila extracts via their interaction with in vitro transcribed RNAs (hairy). Previous work from the Bullock lab has shown that dynein is recruited to these RNAs along with dynactin, Lis1, BicD and Egl. These RNP complexes exhibit a combination of unidirectional movement towards the minus end of the microtubule and bi-directional movement; no unidirectional plus end-directed motility was observed. This work then goes on to characterize these unidirectional and bidirectional types of motility.

Summary recommendations:

The nature of the bidirectional motion can be sorted out in a revision by a) improved MSD analysis, b) higher spatial resolution data of the bidirectional motion, c) analysis of other nucleotide states besides ATP (e.g., ADP/ATP-vanadate or perhaps ADP) and d) most definitive (but perhaps optional) optical trapping to see if the plus end direction is associated with force.

Detailed major concerns:

1) A great deal of this paper involves the analysis and dissection of bidirectional transport, and it seems to me that the interpretation requires an understanding of the principle mechanism of bidirectional motion. The primary question is what is causing motion to the plus end of the microtubule? There is some mention of this issue only at the very end of the discussion section, but it makes it confusing to read along the way and to interpret at the end. Addressing this issue is important for making conclusions such as a ”tug of war” between opposing motors. There are three possibilities:

1) That there is a kinesin moving toward the plus end. This is what most people would assume when “plus end motion or tug of war” is discussed. However, there is no evidence for this in the Results, and in the Discussion, the authors think that this is not the case.

2) That dynein itself is a plus end directed motor, i.e., that it uses ATP energy to execute a prolonged sequence of plus end direct steps (for example a 500 nm run would be about 60 x 8 nm steps).

3) That dynein binds weakly to the microtubule and that the bidirectional motion is thermally driven (perhaps with some mild bias toward the minus end).

This is an important issue and I think that the authors need to confront it in a head-on and convincing way for publication in *eLife*. There are some papers in the literature that suggest model 2 (dynein can either be minus or plus end-directed motor, and perhaps can convert between the two states). This would be a significant discovery if true, but convincing evidence is needed to advance the field. It is important to rule out diffusion as a mechanism; an MSD plot for one run can be misleading since a single run with limited events can produce a deviation for diffusion which could look like “active transport”. Thus, this is not an easy question to answer in a convincing way. Simulation of 1-D diffusion (to see how often if produces apparent unidirectional runs) might be helpful. But better data would be most helpful. Looking at stepping behavior directly at higher resolution might be helpful. The 6 GFP RNPs should be pretty good for analyzing stepping behavior of the RNPs (how unidirectional is the motion when one gets down to <10 nm resolution?). It might also be possible to attach the RNP to a Q dot. Second, is the motion in the plus end direction associated with significant force production? This could be done by attaching the RNP to a 1μm bead. Is the plus end direction associated with a ∼6 pN force (this might be suggestive of a kinesin, but would not rule out a dynein or multiple dyneins; it would rule out diffusion)? Even a 1-2 pN force (what has been reported for a single dynein) might rule that out diffusion. It would also be powerful to show that the plus end force (like the unidirectional minus end) scales with copy number of dynein. However, if there is little force, then this would suggest a diffusive model. Although less informative than the two prior suggestions, it could be interesting to add ATP-vanadate to kill unidirectional minus end movement and see what happens to the “bidirectional” RNPs – is the plus end direction affected? If not, then this might be diffusional. The authors might have other ideas on how to address this issue as well. The bottom line is that the authors need to better support their explanation of an active plus end-directed motor activity. This might reveal that bidirectional motion is due to a switching behavior of the dynein motor itself. If shown convincingly, this would be very interesting and would increase the interest in the paper considerably. It could then be featured in the abstract for sure.

It is clear from the shown data that two types of runs are observed – unidirectional runs and runs where frequent switching of direction takes place. A lot of the conclusions are based on the MSD analysis of individual runs shown in Figure 1—figure supplement 1. This analysis is thus of crucial importance for the interpretation of the rest of the data in this paper and should therefore be moved, together with the kymographs, to the main figures.

It is unfortunate that each MSD plot is obtained from just a single event. In such cases, the calculated values at higher time lags become increasingly inaccurate, because they are obtained by averaging much less displacements than at shorter time lags. In these cases, it is recommended (and common practice) to only analyse the MSD trace until the time lag that corresponds to one quarter of the total trace length. (see Saxton, Biophys J 72, 1744-1753, 1997). Therefore, the ranges that were used to fit the MSD curves seem inappropriate, as is the interpretation of the inset shown in Figure 1—figure supplement 1.

In addition, rather than analysing just single events to obtain an MSD trace, it is common practice to analyse many more events and analyse the data in some statistically correct manner. Since many proteins have been observed to perform diffusive motility along microtubules (e.g., MCAK, Eg5, Ase1, PRC1), several procedures have been firmly established for this.

The authors observe back-and-forth movements along microtubules. The essential question is whether this motility is active, i.e., driven by ATP-dependent molecular motors, or passive, i.e., driven by thermal excitation. The apyrase experiment does not solve this question, as it could lock the motor into a tightly bound state. Additional experiments should include the use of ADP.

The authors state that: “MSD analysis of individual bidirectional RNPs suggested that there was both a diffusive and a deterministic component to their movement in each direction along the microtubule. These observations imply that individual bidirectional RNA–motor complexes undergo directed transport interspersed with diffusive motion.” These conclusions were based on analyzing the complete MSD trace of an individual trajectory, which is incorrect. In addition, the reasoning suggests that any quadratic dependence somewhere in the MSD trace is evidence for deterministic motion (as reported in Figure 1—figure supplement 1). This also seems incorrect. In the case of unidirectional motility, one would indeed expect a quadratic dependence of MSD on time lag, as shown in Figure 1—figure supplement 1. However, for bidirectional motility, there are three different options:

a) Passive, ATP-independent, one-dimensional diffusion on the MT lattice: In this case, one would expect a linear dependence of MSD on time lag for all times with in the range appropriate for analysis (up to one quarter of the total time, when individual traces are analyzed).

b) Active, ATP-dependent bidirectional motility occurring as short bursts of directional motion of average duration “tburst” followed by reversal of directionality. In this case, the MSD traces would appear quadratic/ballistic/deterministic for time lags shorter than “tburst”, and linear for times greater than tburst. In Figure 2, the authors report bursts of ∼500 nm at 1.5 μm/s, suggesting that tburst is approximately 0.3 seconds. Therefore, MSD curves should only be quadratic for times below 0.3 seconds and linear above this.

c) Passive, ATP-independent, one-dimensional diffusion interspersed with occasional (short) runs in the minus end direction every x seconds. In this case, the MSD trace would appear linear up to t = x seconds and become more quadratic at longer times due to the bias introduced by the occasional unidirectional runs. However, also in this case MSD traces from individual trajectories are only reliable up to one quarter of the total time.

Finally, in the description of the plots, the authors state that the red curves represent the deterministic contribution (v2t2) and the black lines the diffusive contribution (2Dt) of the fit to the data (open circles), but this description seems incorrect given that the sum of both contributions is much higher than the real data.

Given these concerns, it is still unclear if bidirectional motility is driven by ATP-dependent motor activity. Even for pure diffusion, the design of subsequent data analysis for Figure 2 will result in separation of the trajectories in apparent persistent runs. Therefore, as it stands, none of these data can be properly interpreted. However, the extreme similarity between plus and minus end 'runs' could be evidence for passive diffusion in both cases.

MSD analysis must therefore be strongly improved.

a) Much more traces should be analyzed and the analysis should be corrected. One should either just average squared trajectories without internal averaging (such as in Helenius et al. Nature 2005) or average MSD traces obtained by internal averaging until one quarter of the total trajectory time.

b) In the case of bidirectional trajectories, testing for a quadratic dependence for MSD(t) as evidence for ATP-driven directional bursts should only be done in the range from 0 to 0.3 seconds. Plotting the MSD(t) on a log-log scale would be helpful for such a test.

c) Nucleotide dependence should also be tested by using ADP instead of ATP.

2) What is the relative amount of dynein/dynactin recruited to HLE compared to *h*^*wt*^, *h*^*ΔLE*^, *h*^*SL1x3*^? This information seems essential for interpreting the experiments in Figure 5. Overall, I found the HLE experiments confusing. Are Egl and BicD present on HLE? What is the basis for dynein/dynactin recruitment to HLE RNA?

Perhaps one could examine the effect of transport of RNPs by changing the copy number of the HLE (e.g., a HLE-short oligo-HLE-short oligo-HLE). Looking at the data, the HLE seems to be a cleaner recruiter of active minus end-directed dynein and could be good for looking for effects of what happens with increasing number of active dyneins. The whole hairy mRNA is more physiological but it seems to recruit some “unidirectional and bidirectional” dyneins, which might complicate the interpretation of how increasing copy number affects motor properties. At minimum, it would be interesting to have compare 1x and 3x HLE and see if the same conclusions (e.g., processivity, velocity, mean run time, reversals, etc) are true as for 1x and 3x hairy mRNA.

---

## [Author Response]

*Summary*
*recommendations:*

*The nature of the bidirectional motion can be sorted out in a revision by a) improved MSD analysis, b) higher spatial resolution data of the bidirectional motion, c) analysis of other nucleotide states besides ATP (e.g., ADP/ATP-vanadate or perhaps ADP) and d) most definitive (but perhaps optional) optical trapping to see if the plus end direction is associated with force*.

Although we initially intended to address this issue in another study, we now appreciate that it is important to tackle it in the current manuscript. The suggested experiment of using ATP-vanadate, which inhibits dynein’s ATP hydrolysis and mimics the ADP.Pi state, was extremely informative. The outcome of this experiment showed that whereas the unidirectional minus end-directed movements depend on an active energy transducing process of dynein, the bidirectional RNPs do not (Figure 2; Figure 2—figure supplement 1 and Figure 2—figure supplement 2). Whilst we previously favoured bidirectional RNP movement representing diffusive movement of dynein-dynactin interspersed with bouts of active transport in each direction (as reported for individual mammalian dynein-dynactin complexes by [59], Nat Cell Biol (PMID 16715075)), our new data indicate that even the longest movements of bidirectional RNPs are due to passive diffusion. Our improved MSD analyses, performed following recommendations in the review (Figure 2 and Figure 2—figure supplement 1 (see below for details)), are also consistent with bidirectional motion being driven by diffusion.

We believe that, in light of the outcome of the vanadate experiments and improved MSD analysis, the suggested experiments of performing force measurements of individual RNPs or visualising stepping patterns with a fluorescence-based method such as FIONA are unnecessary. In any case, both these kinds of experiments would be extremely challenging and therefore not possible within a reasonable timeframe. For example, visualising RNP motility using fluorescence microscopy with enough precision to be able to resolve individual steps will probably require direct labelling of dynein with fluorophores, rather than indirect labelling of the cargo. Developing a method for unobtrusive labelling of dynein is of course time-consuming in a system that deals with native motor complexes from an animal model. There would also be considerable complexity associated with resolving the position of individual dyneins within a multi-dynein cargo-motor assembly. Optical trap experiments would require attaching a large bead to individual RNPs, which (a) is technically not trivial and (b) would not be physiological.

The requested photobleaching analysis of dynein copy number on the *h* localisation element (Figure 4 and Figure 4—figure supplement 1) has also been performed and the results strengthen substantially the evidence for our model that net motion of RNPs is controlled by regulation of dynein activity, rather than total copy number of the motor.

*Detailed*
*major concerns:*

*1) A great deal of this paper involves the analysis and dissection of bidirectional transport, and it seems to me that the interpretation requires an understanding of the principle mechanism of bidirectional motion. The primary question is what is causing motion to the plus end of the microtubule? There is some mention of this issue only at the very end of the discussion section, but it makes it confusing to read along the way and to interpret at the end. Addressing this issue is important for making conclusions such as a “tug of war” between opposing motors*.

We have now expanded Figure 2 to include further experiments and analysis of the nature of bidirectional RNP motion (see below). Consequently, this part of the Results section has changed significantly, with the reader now informed of our key conclusions at that stage. As a consequence of our new results, the discussion of “tug-of-war” has been removed from the manuscript.

*There are three*
*possibilities:*

*1) That there is a kinesin moving toward the plus end. This is what most people would assume when “plus end motion or tug of war” is discussed. However, there is no evidence for this in the Results, and in the Discussion, the authors think that this is not the case*.

*2) That dynein itself is a plus end directed motor, i.e., that it uses ATP energy to execute a prolonged sequence of plus end direct steps (for example a 500 nm run would be about 60 x 8 nm steps)*.

*3) That dynein binds weakly to the microtubule and that the bidirectional motion is thermally driven (perhaps with some mild bias toward the minus end)*.

*This is an important issue and I think that the authors need to confront it in a head-on and convincing way for publication in* eLife*. There are some papers in the literature that suggest model 2 (dynein can either be minus or plus end-directed motor, and perhaps can convert between the two states). This would be a significant discovery if true, but convincing evidence is needed to advance the field. It is important to rule out diffusion as a mechanism; an MSD plot for one run can be misleading since a single run with limited events can produce a deviation for diffusion which could look like “active transport”. Thus, this is not an easy question to answer in a convincing way. Simulation of 1-D diffusion (to see how often if produces apparent unidirectional runs) might be helpful. But better data would be most helpful. Looking at stepping behavior directly at higher resolution might be helpful. The 6 GFP RNPs should be pretty good for analyzing stepping behavior of the RNPs (how unidirectional is the motion when one gets down to <10 nm resolution?). It might also be possible to attach the RNP to a Q dot. Second, is the motion in the plus end direction associated with significant force production? This could be done by attaching the RNP to a 1μm bead. Is the plus end direction associated with a ∼6 pN force (this might be suggestive of a kinesin, but would not rule out a dynein or multiple dyneins; it would rule out diffusion)? Even a 1-2 pN force (what has been reported for a single dynein) might rule that out diffusion. It would also be powerful to show that the plus end force (like the unidirectional minus end) scales with copy number of dynein. However, if there is little force, then this would suggest a diffusive model. Although less informative than the two prior suggestions, it could be interesting to add ATP-vanadate to kill unidirectional minus end movement and see what happens to the “bidirectional” RNPs – is the plus end direction affected? If not, then this might be diffusional. The authors might have other ideas on how to address this issue as well*.

As summarised above, our new data with ATP-vanadate provides compelling evidence in support of bidirectional RNP motion being principally driven by passive diffusion. Specifically, we find that whereas unidirectional RNP movements are abolished by ATP-vanadate, the percentage of microtubule-associated RNPs that are bidirectional is not decreased (Figure 2). Similar distributions of lengths and velocities or individual runs in both the minus and plus end direction were also observed for bidirectional RNPs in both conditions (Figure 2 and Figure 2—figure supplement 2). In fact, we observed a subtle (but statistically significant) increase in run lengths in both directions in the presence of vanadate, suggesting a modulation of diffusive properties of RNPs when dynein is in the ADP.Pi state. Thus, we found no evidence that bidirectional RNPs can undergo ATP hydrolysis-dependent runs in both directions..

*The bottom line is that the authors need to better support their explanation of an active plus end-directed motor activity. This might reveal that bidirectional motion is due to a switching behavior of the dynein motor itself. If shown convincingly, this would be very interesting and would increase the interest in the paper considerably. It could then be featured in the abstract for sure*.

We previously stated that “These observations imply that individual bidirectional RNA–motor complexes undergo directed transport interspersed with diffusive motion”. We are very pleased that the reviewers’ comments led us to refine our understanding of the basis of back-and-forth RNP motion. However, as included in a letter to Prof. Schekman after the initial reviews, we think that a scenario in which bidirectional RNPs exclusively undergo passive diffusion along the microtubule is no less interesting than our previous model. This is particularly the case given recent evidence in *S. pombe* that dynein can undergo diffusive motion along microtubules in vivo (Ananthanarayanan et al. Cell, 2013. PMID 23791180). In fact, our new data substantially clarify the interpretation of our whole study. They lead to a novel model in which the same cargo species can associate with processive or non-processive dynein, with discrete cargo-associated features controlling switching between the two behaviours. This work therefore also paves the way for identifying factors and mechanisms that regulate dynein processivity. Our realisation that bidirectional motion is principally driven by diffusion also refines substantially our interpretations of the experiments assessing the behaviour of RNPs at MAPs and microtubule ends.

*It is clear from the shown data that two types of runs are observed – unidirectional runs and runs where frequent switching of direction takes place. A lot of the conclusions are based on the MSD analysis of individual runs shown in*
Figure 1—figure supplement 1*. This analysis is thus of crucial importance for the interpretation of the rest of the data in this paper and should therefore be moved, together with the kymographs, to the main figures*.

We agree with the reviewers. We have now included improved MSD data for *h*^*WT*^ RNPs that are unidirectional or bidirectional in Figure 2. We also show examples of kymographs in Figure 1. We have also added supplementary data from the new MSD analysis in Figure 2—figure supplement 1 (see below for more details).

*It is unfortunate that each MSD plot is obtained from just a single event. In such cases, the calculated values at higher time lags become increasingly inaccurate, because they are obtained by averaging much less displacements than at shorter time lags. In these cases, it is recommended (and common practice) to only analyze the MSD trace until the time lag that corresponds to one quarter of the total trace length. (see Saxton, Biophys J 72, 1744-1753, 1997). Therefore, the ranges that were used to fit the MSD curves seem inappropriate, as is the interpretation of the inset shown in*
Figure 1—figure supplement 1.

We are very grateful to the reviewer for their substantial effort to help us correct the analysis. MSD analysis of individual RNPs (Figure 1—figure supplement 1) is now for the time period that corresponds to one quarter of the total trace length. This information is described in the relevant figure legend and in the methods.

*In addition, rather than analyzing just single events to obtain an MSD trace, it is common practice to analyze many more events and analyze the data in some statistically correct manner. Since many proteins have been observed to perform diffusive motility along microtubules (e.g., MCAK, Eg5, Ase1, PRC1), several procedures have been firmly established for this*.

We now present data in Figure 2 and Figure 2—figure supplement 1 that is averaged from many individual RNP trajectories. In this analysis, the MSD analysis is only performed for the time corresponding to one quarter of the duration of the RNP trajectory with the shortest lifetime.

*The authors observe back-and-forth movements along microtubules. The essential question is whether this motility is active, i.e., driven by ATP-dependent molecular motors, or passive, i.e., driven by thermal excitation. The apyrase experiment does not solve this question, as it could lock the motor into a tightly bound state. Additional experiments should include the use of ADP*.

We assume that here the reviewers are referring to the ADP.Pi state of the motor, which is mimicked by ADP-vanadate. ADP would also be expected to lock dynein in a tightly bound state (e.g., Holzbaur and Johnson, Biochemistry 1989, PMID 2531005; [59]), and therefore the interpretation of such an experiment would suffer from the same caveat as the apyrase experiment.

*The authors state that: “MSD analysis of individual bidirectional RNPs suggested that there was both a diffusive and a deterministic component to their movement in each direction along the microtubule. These observations imply that individual bidirectional RNA–motor complexes undergo directed transport interspersed with diffusive motion.” These conclusions were based on analyzing the complete MSD trace of an individual trajectory, which is incorrect. In addition, the reasoning suggests that any quadratic dependence somewhere in the MSD trace is evidence for deterministic motion (as reported in*
Figure 1—figure supplement 1*). This also seems incorrect. In the case of unidirectional motility, one would indeed expect a quadratic dependence of MSD on time lag, as shown in*
Figure 1—figure supplement 1*. However, for bidirectional motility, there are three*
*different options:*

*a) Passive, ATP-independent, one-dimensional diffusion on the MT lattice: In this case, one would expect a linear dependence of MSD on time lag for all times with in the range appropriate for analysis (up to one quarter of the total time, when individual traces are analyzed)*.

*b) Active, ATP-dependent bidirectional motility occurring as short bursts of directional motion of average duration “tburst” followed by reversal of directionality. In this case, the MSD traces would appear quadratic/ballistic/deterministic for time lags shorter than “tburst”, and linear for times greater than tburst. In*
Figure 2*, the authors report bursts of ∼500 nm at 1.5 μm/s, suggesting that tburst is approximately 0.3 seconds. Therefore, MSD curves should only be quadratic for times below 0.3 seconds and linear above this*.

*c) Passive, ATP-independent, one-dimensional diffusion interspersed with occasional (short) runs in the minus end direction every x seconds. In this case, the MSD trace would appear linear up to t = x seconds and become more quadratic at longer times due to the bias introduced by the occasional unidirectional runs. However, also in this case MSD traces from individual trajectories are only reliable up to one quarter of the total time*.

Our new MSD analysis supports option “a” (passive, ATP-independent, one-dimensional diffusion on the microtubule lattice). We have averaged squared trajectories from 40 bidirectional RNPs (for the time corresponding to one quarter of the duration of the RNP trajectory with the shortest lifetime). The analysis reveals a linear dependence of MSD on time. These data are documented in Figure 2 (in a log-log plot, which saves on space in this busy figure by allowing unidirectional and bidirectional RNPs to be plotted on the same graph) and Figure 2—figure supplement 1 (without the log-log conversion). We also represent the data for individual RNPs in Figure 2—figure supplement 1 (with a time interval for each RNP that corresponds to one quarter of the total duration of its trajectory). We find this scatter plot informative as it provides evidence against a subset of RNPs being affected by ATP-vanadate and therefore containing significant bouts of active motion.

*Finally, in the description of the plots, the authors state that the red curves represent the deterministic contribution (v2t2) and the black lines the diffusive contribution (2Dt) of the fit to the data (open circles), but this description seems incorrect given that the sum of both contributions is much higher than the real data*.

The analysis of individual RNP trajectories in this manner has been removed from the revised manuscript.

*Given these concerns, it is still unclear if bidirectional motility is driven by ATP-dependent motor activity. Even for pure diffusion, the design of subsequent data analysis for*
Figure 2
*will result in separation of the trajectories in apparent persistent runs*.

This is yet another very important point. Given the distribution of run lengths in the presence and absence of vanadate (Figure 2—figure supplement 2), it is not appropriate to apply a cut-off to the run length analysis. The new manuscript therefore includes data from reanalysis of runs without the cut-off.

*Therefore, as it stands, none of these data can be properly interpreted. However, the extreme similarity between plus and minus end 'runs' could be evidence for passive diffusion in both cases*.

This result is indeed consistent with our data suggesting passive diffusion as the dominant mechanism for bidirectional RNP movement, a point that is now made in the Results section entitled “Characterisation of the motile properties of h^WT^ RNPs” of the revised manuscript: “Both these scenarios, or a combination of the two, could account for the relatively short mean run lengths, diffusive MSD properties and the tight coupling of minus and plus end motile properties for individual bidirectional RNPs in the presence of ATP”.

*MSD analysis must therefore be strongly improved*.

*a) Much more traces should be analyzed and the analysis should be corrected. One should either just average squared trajectories without internal averaging (such as in Helenius et al. Nature 2005) or average MSD traces obtained by internal averaging until one quarter of the total trajectory time*.

As described above, we have taken up the latter suggestion, which appears to be the most commonly used method in the literature.

*b) In the case of bidirectional trajectories, testing for a quadratic dependence for MSD(t) as evidence for ATP-driven directional bursts should only be done in the range from 0 to 0.3 seconds. Plotting the MSD(t) on a log-log scale would be helpful for such a test*.

We have specifically tested the range only from 0 to 0.3 s and find a linear relationship of MSD with time, including when the data are plotted on a log-log scale. In fact we find a linear dependence of MSD on time for bidirectional RNPs for all timescales analysed. Given that the recommendation of the time interval of 0 to 0.3 seconds was based on our previous cut-off for runs of 250 nm – which we have now discarded – we assume that it is not necessary to include this specific analysis in the paper (although we mention the result as data not shown in the legend to Figure 2—figure supplement 1). We would of course be willing to take further guidance from the reviewer on this matter.

*c) Nucleotide dependence should also be tested by using ADP instead of ATP*.

As described above, we have tested the ADP.Pi condition using ATP-vanadate.

*2) What is the relative amount of dynein/dynactin recruited to HLE compared to h*^*wt*^*, h*^*ΔLE*^*, h*^*SL1x3*^*? This information seems essential for interpreting the experiments in*
Figure 5.

We have now performed GFP-Dlic photobleaching analysis on the *HLE* (Figure 4 and Figure 4—figure supplement 1) to confirm that this RNA has a significantly lower copy number of dynein sub-units than the *h*^*wt*^ RNA. The *h*^*wt*^ dataset in these panels was collected in parallel to the *HLE* dataset.

*Overall, I found the HLE experiments confusing*.

We feel that the inclusion of the photobleaching data, suggested by the reviewers, clarifies the logic of the experiment substantially. We have also set up the rationale for the experiment in more detail in the Results section entitled “The *HLE* alone promotes unidirectional motion”: “Importantly, we first confirmed using stepwise photobleaching with GFP-Dlic extracts that there is a significant reduction in the relative copy number of the motor complex on *HLE* RNPs compared to *h*^*wt*^ RNPs (Figure 4 and Figure 4—figure supplement 1). The average number of GFP::Dlic photobleaching steps on the *HLE* was in fact statistically indistinguishable from that observed for *h*^*ΔLE*^ (2.21 ± 0.15 and 2.14 ± 0.16, respectively; Figure 4 and Figure 3).

If total copy number of the dynein complex is the sole determinant of unidirectional transport, one would expect *HLE* RNPs to be less likely than *h*^*wt*^ RNPs to exhibit this behaviour, possibly exhibiting exclusively bidirectional motion as is the case for *h*^*ΔLE*^”.

We have also simplified Figure 4 to help the reader follow the key conclusions and reviewed one of key experiments in the Discussion section “Insights into the mechanisms of cargo sorting by multi-motor assemblies”.

“However, experiments with the isolated *HLE* signal indicated that total motor number is not the key determinant of the unidirectional mode of RNP movement. The *HLE* alone has a statistically indistinguishable copy number of dynein components to the *h*^*ΔLE*^ RNA, in which the localisation signal has been replaced by a heterologous sequence in the context of the *h* 3’UTR, yet only the former RNA is capable of unidirectional motion. These data suggest that features associated with the RNA signal are sufficient to increase the probability of processive movement of the associated dynein”.

We have added a cartoon figure to further clarify our working model (Figure 7).

*Are Egl and BicD present on HLE? What is the basis for dynein/dynactin recruitment to*
*HLE RNA?*

Egl directly binds RNA localisation signals, including the *HLE* (Dienstbier et al. 2009). Egl also directly binds BicD and these two proteins act as adaptors to dynein-dynactin (Bullock et al., 2006; Dienstbier et al. 2009; Dix et al. 2013). Together with dynein and dynactin components, Lis1 and CLIP-190 (which is not functionally important for mRNA motility), Egl and BicD are the only proteins that are specifically associated with localisation signals using the RNA affinity purification method also employed during the RAT-TRAP procedure (Dix et al. 2013). These data are reviewed in the Introduction, and introduced again in the context of the *HLE* in the section entitled “Increasing dynein-dynactin numbers through RNA localization signals…” in the Results: “The key feature of the HLE is stem-loop 1 (SL1) (Bullock et al. 2003), which recruits dynein- dynactin through the adaptor proteins Egl and BicD (Dienstbier et al. 2009; Dix et al. 2013)”.

*Perhaps one could examine the effect of transport of RNPs by changing the copy number of the HLE (e.g., a HLE-short oligo-HLE-short oligo-HLE). Looking at the data, the HLE seems to be a cleaner recruiter of active minus end-directed dynein and could be good for looking for effects of what happens with increasing number of active dyneins. The whole hairy mRNA is more physiological but it seems to recruit some “unidirectional and bidirectional” dyneins, which might complicate the interpretation of how increasing copy number affects motor properties. At minimum, it would be interesting to have compare 1x and 3x HLE and see if the same conclusions (e.g., processivity, velocity, mean run time, reversals, etc) are true as for 1x and 3x hairy mRNA*.

This is an interesting suggestion, but we do think it is important to focus in the current study on the influence of motor number using the longer RNA because of the greater physiological significance associated with recruitment of dynein-dynactin to non-signal sites. Given the time constraints, we elected to concentrate on the points raised in the summary recommendations.